# Graph Learning Assisted Multi-Objective Integer Programming

**Yaoxin Wu**[1], **Wen Song**[2], **Zhiguang Cao**[3,*], **Jie Zhang**[1],
**Abhishek Gupta**[3] **and Mingyan Simon Lin**[3]

[1]Nanyang Technological University
[2]Institute of Marine Science and Technology, Shandong University, China
[3]Singapore Institute of Manufacturing Technology, A*STAR
wuyaoxin@ntu.edu.sg, wensong@email.sdu.edu.cn, zhiguangcao@outlook.com
zhangj@ntu.edu.sg, {abhishek_gupta, simon_lin}@simtech.a-star.edu.sg

## Abstract

Objective-space decomposition algorithms (ODAs) are widely studied for solving multi-objective integer programs. However, they often encounter difficulties in handling scalarized problems, which could cause infeasibility or repetitive nondominated points and thus induce redundant runtime. To mitigate the issue, we present a graph neural network (GNN) based method to learn the reduction rule in the ODA. We formulate the algorithmic procedure of generic ODAs as a Markov decision process, and parameterize the policy (reduction rule) with a novel two-stage GNN to fuse information from variables, constraints and especially objectives for better state representation. We train our model with imitation learning and deploy it on a state-of-the-art ODA. Results show that our method significantly improves the solving efficiency of the ODA. The learned policy generalizes fairly well to larger problems or more objectives, and the proposed GNN outperforms existing ones for integer programming in terms of test and generalization accuracy.

## 1 Introduction

Practical combinatorial optimization often involves multiple objectives that conflict with each other. Such problems can be termed as multi-objective combinatorial optimization (MOCO), which are widely studied in operations research and computer science [1], and also have broad applications in energy [2], engineering [3], biology [4], etc. In contrast to the single-objective optimization, MOCO centers on searching solutions with nondominated images (points) in the objective space, which constitute the Pareto front. These solutions reflect different preferences on the objectives, so that decision makers could select from them according to practical use. In this paper, we focus on multi-objective integer programming (MOIP), which provides a unified modeling framework for MOCOs, and is convenient in handling complex constraints which could be difficult for heuristic methods [5]. However, MOIP problems are extremely difficult to be solved optimally (i.e. enumerate the whole Pareto front) [6, 7]. Therefore, accelerating the computation of MOIPs to ameliorate the partial Pareto front in reasonable time is a critical and challenging issue [8, 9, 10].

Recently, deep learning has been extensively explored in solving single-objective combinatorial optimization problems such as vehicle routing, scheduling, and bin packing [11, 12, 13, 14, 15, 16, 17, 18]. As a general modeling and solving framework, single-objective integer programming (IP) has also received increasing attention, and this line of works mostly attempt to improve ingredients in the branch and bound (BnB) paradigm [19, 20, 21, 22, 23, 24]. In contrast, research on applying deep

---

*Corresponding Author.

36th Conference on Neural Information Processing Systems (NeurIPS 2022).

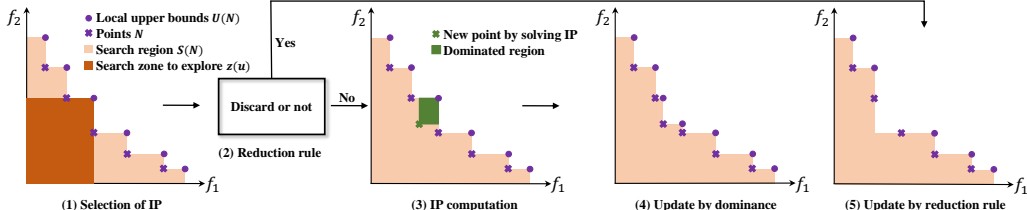

Figure 1: **An example of ODA for solving a MOIP with two objectives.** The algorithmic procedure iterates the following phases, 1) selecting one from all local upper bounds and defining the corresponding scalarized problem (IP); 2) determining whether the IP is to be solved, if Yes 3) solving the IP by calling a solver; 4) updating the search region according to the dominance of the newly derived image (point), if No 5) discarding the selected local upper bound without calling a solver.

learning to MOCO is relatively sparse, with only recent trials targeting at specific problems such as travelling salesman and knapsack problem [10, 25, 26, 27] rather than the more general MOIP.

This paper, for the first time, presents a graph learning based method to assist MOIP solving by improving the bounded-time performance of objective-space decomposition algorithms (ODAs). With highly optimized IP solvers such as CPLEX and Gurobi, ODAs iteratively decompose a MOIP into multiple scalarized problems (i.e. IPs) by applying local upper bounds in the objective space and solve one of the IPs to update the bounds, as illustrated in Figure 1. Since ODAs involve computation primarily in the low-dimensional objective space which could be fulfilled by calling mature IP solvers, this paradigm has attracted much research in the literature [8, 28, 29, 30]. However, existing ODAs often solve large amounts of IPs that 1) are *infeasible* or 2) lead to *repetition* of nondominated points, which result in unnecessary and redundant computation.

To tackle this issue, we propose to learn reduction rules to discard local upper bounds that induce futile IPs pertaining to infeasibility or repetition in iterations of ODAs, so as to reduce unnecessary computation eventually. To this end, we model the algorithmic procedure of ODAs as a Markov decision process based on the analysis of classic reduction rules in ODAs. To resolve the challenges in representing MOIP solving states, we propose a novel two-stage graph neural network (GNN) to parameterize the policy (reduction rule). In the first stage, the variable, constraint and objective nodes attend to each other locally to update their embeddings; while in the second stage, the embeddings are aggregated into three hypernodes for each pair of node types to realize a better global representation. We train the GNN based policy with imitation learning so that it learns to discard local upper bounds of futile IPs. We employ our method to improve a state-of-the-art ODA [29], and extensive results on benchmark problems indicate its efficacy in accelerating the search of nondominated points in bounded time and generalizing to larger problems or more objectives. We also compare the proposed GNN with existing ones for IPs to verify its superiority in learning representations for MOIPs.

## 2 Related work

### 2.1 Objective-space decomposition for MOIP

A variety of methods have been proposed for MOIPs. Among them, ODAs attract much attention since they usually resort to highly-optimized IP solvers for solving the decomposed scalarized problems, with additional light computation only processing local bounds in a low-dimensional objective space. IP solutions to scalarized problems update local bounds to delimit a search region where nondominated points exactly exist. As representatives, Kirlik and Sayın [31] equip $\epsilon$-constraint with $p$-1 dimensional boxes representing the search region, and iteratively split them to filter out the ones without nondominated points. Klamroth et al. [28] formally describe the search region with full dimensional zones whose amount is significantly less than that of boxes in [31], inducing less subroutines for IPs. Dächert et al. [32] accelerate the update of the search region with a specific neighborhood relation between local upper bounds. Boland et al. [33] define the search region by known efficient solutions and extra disjunctive constraints, which increased the computational complexity in IP solving. Tamby and Vanderpooten [29] point out the infeasibility issue and propose a state-of-the-art framework for MOIPs where infeasible programs could be avoided. However, this

method needs to restore all solved IPs and look up their feasible domains to check feasibility of the current IP in each iteration. Moreover, existing ODAs generally lack a capacity to detect repetitive IPs. This paper provides a graph learning based method to fast bypass infeasibility and repetition of nondominated points. Other less related works pertaining to ODAs can be found in [34, 35, 30].

## 2.2 Graph learning for integer programs

Graph neural networks (GNNs) have been explored to learn representations for single-objective IPs, which identify problem structures better than trivial neural networks (e.g. multilayer perceptron (MLP)) and enable generalization to different scales. Gasse et al. [21] describe IPs at each node in BnB search as bipartite graphs and use graph convolutional network (GCN) to learn branching rules. It inspires a number of subsequent works to employ miscellaneous GNNs for tackling IPs. Ding et al. [36] propose tripartite graph embedding for IPs to predict values of binary variables. Gupta et al. [37] consider a hybrid use of GCN and MLP to accelerate the branching in BnB. Nair et al. [24] adapt the bipartite GCN in [21] to learn both branching rules and a specific diving heuristic at the root node. Wu et al. [38] apply it to select variables in IPs for destroy operations in a large neighborhood search framework. Recently, Khalil et al. [39] evolve the bipartite GCN by substituting its message passing structure into GIN and GraphSAGE [40, 41], to separately train node selection rules and predict warm starts for BnB search. The existing graph learning based methods might be inadequate to represent MOIPs where multiple objective nodes need to be identified and the potential scalarized problems need to be discriminated. In this paper, we propose an effective graph encoding for MOIPs, and design a novel two-stage GNN to learn the corresponding graph representations. We update node embeddings by attention in objective-variable, variable-constraint graphs, and aggregate these (locally) advanced node embeddings into hypernodes to further gain a favorable graph representation.

# 3 Preliminaries

## 3.1 Multi-objective integer programs

The integer-constrained optimization with $p$ objective functions are formally described as follows:

$$
\begin{aligned}
\min_{x} \quad & f(x) = (f_1(x), \ldots, f_p(x)) \\
\text{s.t.} \quad & Ax \leqslant b, \ x \in \mathbb{Z}^{n_z} \times \mathbb{R}^{n-n_z},
\end{aligned}
\tag{1}
$$

where $A \in \mathbb{R}^{m \times n}$, $b \in \mathbb{R}^m$, $m, n, n_z \in \mathbb{N}$ and $n \geqslant n_z > 0$. The MOIP is specified from the above formulation with $n = n_z$. Given a feasible solution $x$, its image (point) in the objective space is $y = f(x) = (f_1(x), \ldots, f_p(x))$, resulting in the set of feasible solutions $\mathcal{X}$ and their images $\mathcal{Y}$. In $\mathcal{Y}$, an image $y$ is *weakly dominated* by $y'$ (i.e. $y' \leqq y$), if $y_i' \leqslant y_i, \forall i \in \{1, \ldots, p\}$; $y$ is *dominated* by $y'$ (i.e. $y' \leq y$), if $y' \leqq y$ and $y \neq y'$; $y$ is *strictly dominated* by $y'$ (i.e. $y' < y$), if $y_i' < y_i, \forall i \in \{1, \ldots, p\}$. Accordingly, a solution $x \in \mathcal{X}$ is *efficient* or *weakly efficient* if its image $f(x)$ is not dominated or strictly dominated by any other image in the objective space. The common goal of MOIPs is searching the Pareto front that is the set of nondominated images $\mathcal{Y}_{nd} = \{f(x) | x \in \mathcal{X}_e\}$ derived from efficient solutions $\mathcal{X}_e$.

## 3.2 Objective-space decomposition algorithms

Following [28, 32, 29], we describe the search region by local upper bounds to locate the remaining nondominated points at each iteration in ODAs.

**Definition 1.** Given a set of points (images) $N$ in $p$-dimensional objective space, the corresponding search region is defined as $S(N) = \{y \in \mathbb{R}^p | y' \nleqq y, \forall y' \in N; y^I \leqq y\}$, where $y^I$ is the ideal point with $y_i^I = \min_{x \in \mathcal{X}} f_i(x) = \min_{y \in \mathcal{Y}_{nd}} y_i$.

To concisely represent the search region and facilitate its decomposition into scalarized problems, the (maximal) local upper bounds are introduced [28].

**Definition 2.** A set of points $U(N)$ is called (maximal) local upper bounds of $N$ if and only if 1) $S(N) = \bigcup_{u \in U(N)} z(u)$, where $z(u) = \{y \in \mathbb{R}^p | y^I \leqq y < u\}$ is the search zone under local upper bound $u$; 2) $\forall u, u' \in U(N), u \nleqq u'$.

The "maximal" emphasizes the condition 2), i.e., $\forall u, u', z(u) \nsubseteq z(u')$. Given the current set of local upper bounds and a new nondominated point, the search region (with the local upper bounds) can be efficiently updated by excluding the dominated region of the point via the algorithm in [28]. Such update procedure keeps shrinking the search region to locate the remaining nondominated points, and we elaborate it in Appendix A.1.

**Proposition 1.** Given $u \in U(N)$ and a strictly monotone function[2] $\mathcal{F} : \mathcal{Y} \rightarrow \mathbb{R}$, the optimal solution to the following scalarized problem (IP) will derive a nondominated point in $\mathcal{Y}_{nd}$,

$$\Gamma(u) := \min\{\mathcal{F}(y) = \mathcal{F}(y_1, \ldots, y_p) | y \in \mathcal{Y}; y < u\}. \tag{2}$$

The common instantiations of $\mathcal{F}$ include $\epsilon$-constraint, Lexicographic and weighted Tchebycheff methods [42, 8], which are detailed in Appendix A.2. Given the above concepts, the generic algorithmic procedure for ODAs starts from a (local) upper bound $U(\emptyset) = \{(M, \ldots, M)\}$ with $M$ being a large value,[3] and comprises four components, 1) selection rule that picks $u$ from $U(N)$ in Eq. (2); 2) reduction rule that detects feasibility of IP in Eq. (2); 3) IP solving with a solver (e.g. Gurobi, CPLEX); 4) update of search region $U(N)$ according to dominance or reduction rule. This algorithmic paradigm has the property in Proposition 2. We prove Proposition 1, 2 in Appendix A.3.

**Proposition 2.** ODAs generate the whole Pareto front for MOIP with finite iterations.

## 4 Methodology

In this section, we identify the critical issue about the infeasibility and repetition of nondominated points in ODAs, and propose to circumvent the solving of the corresponding IPs with a reduction rule learned automatically by GNN. We encode the generic algorithmic procedure of ODAs as a MDP, where the reduction rule is parameterized by a two-stage GNN and trained with imitation learning.

### 4.1 Problem statement: infeasibility and repetition reduction

In ODAs, most of the computation time in each iteration is consumed by IP solving (normally with mature solvers). However, existing algorithms are less effective in identifying and discarding local upper bounds that induce futile IPs with infeasible or repetitive nondominated points. The commonly used reduction rules rely on the inclusion relationship of solution spaces between IPs solved and to be solved [31, 33, 29]. We summarize this class of reduction rules as follows.

**Observation 1.** Given a scalarized problem $\Gamma(u)$, it is infeasible if there exists a $\Gamma(u')$ with, 1) the same $\mathcal{F}$ as in $\Gamma(u)$, and 2) $u \leqq u'$, which has been solved and returns null.

Since all solutions in $\Gamma(u)$ are subset of those in $\Gamma(u')$, they have been proved infeasible under the same objective function by $\Gamma(u')$. However, reduction rules based on observation 1 require $\Gamma(u')$ to be solved to prove infeasibilty, which induces extra runtime. We extend it to the optimality cases.

**Proposition 3.** Given a scalarized problem $\Gamma(u)$, it is infeasible if there exists a solved $\Gamma(u')$ with 1) the same $\mathcal{F}$ as in $\Gamma(u)$, 2) the optimal point $y^*$ satisfying $\mathcal{F}(u) \leqslant \mathcal{F}(y^*)$, and 3) $u \leqq u'$.

**Proof.** Given $u \leqq u'$, the solution space of $\Gamma(u')$ contains the one of $\Gamma(u)$. Since the optimal point $y^*$ has achieved the lower bound in $\Gamma(u')$, $\Gamma(u)$ cannot attain any feasible solution $y'$ with $\mathcal{F}(y') < \mathcal{F}(u) \leqslant \mathcal{F}(y^*)$ based on the strictly monotone property in Proposition 1. If so, it contradicts with the optimality of $y^*$ in $\Gamma(u')$.

While the instantiation for Observation 1 as the reduction rule can be found in [31, 33], Proposition 3 is used as a reduction rule with Lexicographic scalarization in [29]. However, the above methods need clearly more memory and extra runtime to trace or iterate over the IPs to check infeasibility. Meanwhile, the repetition of nondominated points also broadly exists in ODAs, where existing methods generally detect them by calling solvers, inducing extra runtime as well [31, 29, 30].

To curtail unnecessary solver calls that cause infeasibility and repetition, we incline to fast predict these status immediately after a local upper bound is selected with its IP formulation. In the following, we convert the algorithmic procedure of ODAs into a MDP, and learn a reduction rule by GNN to discard the local upper bounds that may induce infeasible and repetitive points in corresponding IPs.

---

[2]A strictly monotone function ensures that: $\forall y, y' \in \mathbb{R}^p$, if $y \leq y'$, then $\mathcal{F}(y) < \mathcal{F}(y')$.
[3]In practice, $M$ could be provided by a decision maker to constrain the search region of interest [28].

**Remark.** Since we learn the reduction rule to accelerate the detection of unnecessary IPs, an undesirable byproduct is that some IPs with effective nondominated points could be ignored, which may impair the completeness of the found Pareto points. However, our learned reduction rule has favorable potential to achieve a good trade-off that it is able to search reasonably more nondominated points in the bounded time, especially for large-scale problems.

## 4.2 Markov decision process

For each MOIP instance, we describe the solving process of ODAs as a MDP, where the reduction rule is considered as the agent and remaining algorithmic procedures as the environment. At the $t$-th iteration, the state $s_t$ is characterized by the current algorithmic status including the fixed information of the given MOIP instance; the current nondominated points $N_t$ and their statistics; the found efficient solutions $\mathcal{X}_e^t$ and their statistics; the set of local upper bounds and the selected one $u_t$. The agent determines whether the selected local upper bound $u_t$ is discarded (1) or not (0), i.e., $a_t \in \mathcal{A} = \{0, 1\}$, via the learned policy $\pi(a_t|s_t)$. With the decision from the agent, the environment transits from $s_t$ to $s_{t+1}$ by either solving the IP defined by $u_t$ and updating the search region $S(N)$, or directly discarding $u_t$ from $U(N)$. The above one-step transition iterates to yield an episode $\tau = (s_0, a_o, s_1, \ldots, s_T)$ with its probability $p(\tau) = p(s_0) \prod_{t=0}^{T-1} \sum_{a_t} p(s_{t+1}|s_t, a_t)\pi(a_t|s_t)$. According to Proposition 2, the length of the trajectory (episode) is finite and generally larger than the cardinality of Pareto front due to the presence of futile IPs. The goal of MDP is to learn an optimal policy defined as below.

**Definition 3.** The optimal policy $\pi^*$ for an ODA is the one finalizing, 1) $N_T$ with all points in the Pareto front, and 2) the frequency of $a_t = 0$ in $\tau$ always equal to the cardinality of the Pareto front.

In other words, it would be ideal that with the learned reduction rule, every subroutine of the solver can produce one (effective) nondominated point, and all local upper bounds with their IPs resulting in infeasible or repetitive points are discarded with $\pi^*(a_t = 1|s_t) = 1$.

## 4.3 Graph learning

For graph encoding of states $\{s_t\}_{t=0}^{T-1}$ in the above MDP, it is crucial to represent multiple objective nodes for each respective MOIP as in Eq. (1) and identify varying scalarized problems (IPs) as in Eq. (2) along the solving process. To this end, we define a graph $\mathcal{G} = (\mathcal{C}, \mathcal{V}, \mathcal{O}, \mathcal{E}_{cv}, \mathcal{E}_{ov})$ with constraint nodes $\mathcal{C} = \{c_1, \cdots, c_m, c_{m+1}, \cdots, c_{m+p}\}$ featurized by a matrix $\mathbf{C} \in \mathbb{R}^{(m+p) \times d_c}$, variable nodes $\mathcal{V} = \{v_1, \cdots, v_n\}$ featurized by $\mathbf{V} \in \mathbb{R}^{n \times d_v}$, objective nodes $\mathcal{O} = \{o_1, \cdots, o_p\}$ featurized by $\mathbf{O} \in \mathbb{R}^{p \times d_o}$, edges $\mathcal{E}_{cv} = \{e_{ij}|c_i \in \mathcal{C}, v_j \in \mathcal{V}\}$ between constraint and variable nodes featurized by $\mathbf{E}_{cv} \in \mathbb{R}^{(m+p) \times n \times d_{cv}}$, edges $\mathcal{E}_{ov} = \{e_{kj}|o_k \in \mathcal{O}, v_j \in \mathcal{V}\}$ between objective and variable nodes featurized by $\mathbf{E}_{ov} \in \mathbb{R}^{p \times n \times d_{ov}}$. To derive accurate global representations of states, we further attach three hypernodes $\mathcal{H} = \{h_1, h_2, h_3\}$ to aggregate node representations from three partial graphs, that is, $\mathcal{G}_1 = (\mathcal{C}, \mathcal{V})$, $\mathcal{G}_2 = (\mathcal{O}, \mathcal{V})$ and $\mathcal{G}_3 = (\mathcal{C}, \mathcal{O})$. We illustrate the above graph in Figure 2 (the left one). The proposed graph encoding is generic for MOIPs, in which we can capture different scalarized problems by featurization (e.g. updating $u$ for each IP). In this paper, we realize the featurization with respect to the Lexicographic scalarizations for the ODA [29], and the details are provided in Appendix A.4.

**Two-stage GNN.** Given the graph encoding, we propose a two-stage GNN (illustrated in the right half of Figure 2) to parameterize the reduction rule as $\pi_\theta(a_t|s_t)$. It pertains to the class of message-passing neural networks [43] with graph attention structure [44] and is enhanced by hypernode embedding which aggregates high-level graph information. In the first stage, the total four passes are executed along the two groups of undirected edges in the order $\mathcal{V} \to \mathcal{C} \to \mathcal{V} \to \mathcal{O} \to \mathcal{V}$, where the representations of target nodes for each pass are advanced as follows (taking $\mathcal{V} \to \mathcal{C}$ as an example),

$$\alpha_{ij} = \frac{\text{SUM}\left(\text{LR}\left(\mathbf{v}_j W_v + \mathbf{e}_{ij} W_{cv} + \mathbf{c}_i W_c\right) \cdot W_a\right)}{\sum_{v_j \in \mathcal{V}} \text{SUM}\left(\text{LR}\left(\mathbf{v}_j W_v + \mathbf{e}_{ij} W_{cv} + \mathbf{c}_i W_c\right) \cdot W_a\right)},$$

$$\mathbf{c}_i = \text{MLP}_1((1 + \varepsilon) \cdot \mathbf{c}_i + \text{MLP}_2(\sum_{v_j \in \mathcal{V}} \alpha_{ij} \cdot \mathbf{v}_j)), \quad \forall i, c_i \in \mathcal{C} \tag{3}$$

where $\mathbf{c}_i$, $\mathbf{v}_j$ denote the $i$-th and $j$-th row of matrice $\mathbf{C}$ and $\mathbf{V}$ (i.e. features of $c_i$ and $v_j$), and are separately projected into $d_r$-dimensional embeddings before feeding them into Eq. (3); $W_v, W_{cv}, W_c, W_a \in \mathbb{R}^{d_r \times d_r}$ are trainable parameters to compute the attention weight $\alpha_{ij}$; LR and

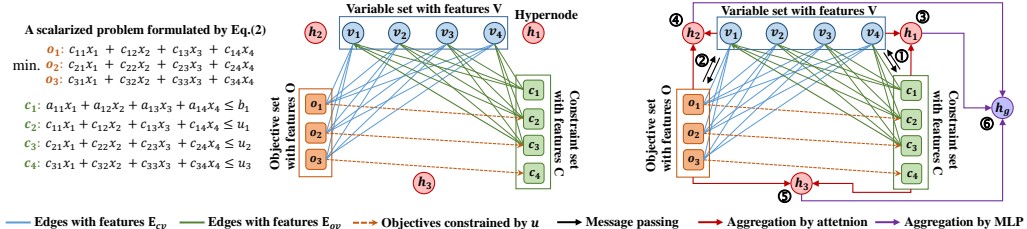

Figure 2: **An illustration of graph learning for MOIP instances with three objectives and one constraint.** The left subfigure shows the graph encoding for scalarized problems (IPs) in Eq. (2), where three constraints are varying since objectives are constrained by different local upper bound $u$ along the solving process. The right subfigure shows the message flow with circled numbers, where ① and ② correspond to four message passes in the first stage among objectives, variables and constraints; ③ ∼ ⑥ correspond to the two-step aggregation in the second stage.

SUM mean Leaky ReLU activation and the addition of elements, respectively; $\text{MLP}_1$ and $\text{MLP}_2$ are two multilayer perceptrons, and each has two linear layers and one ReLU activation; $\varepsilon$ could be a trainable parameter [40] and we set it to 0 in this paper. Following the above structure and logic, the representation of all nodes are updated after all passes are executed.

Since we aim to learn prediction for IPs that are encoded on graphs, an accurate representation of the entire graph is the foremost concern. Hence, in the second stage, we propose to aggregate the graph representation in a two-step manner as expressed in Eq. (4). Specifically, we first aggregate partial graphs $(\mathcal{G}_1, \mathcal{G}_2, \mathcal{G}_3)$ on hypernodes $(h_1, h_2, h_3)$ with attentions and then aggregate those hypernodes to attain the graph representation,

$$
\begin{aligned}
&\mathbf{h}_1, \mathbf{h}_2, \mathbf{h}_3 = \text{Softmax}(\text{MLP}_3(\mathbf{H}_1))\mathbf{H}_1, \text{Softmax}(\text{MLP}_4(\mathbf{H}_2))\mathbf{H}_2, \text{Softmax}(\text{MLP}_5(\mathbf{H}_3))\mathbf{H}_3, \\
&h_g = \text{MLP}_6([\mathbf{h}_1; \mathbf{h}_2; \mathbf{h}_3]), \quad \text{given } \mathbf{H}_1 = [\mathbf{C}; \mathbf{V}], \mathbf{H}_2 = [\mathbf{O}; \mathbf{V}], \mathbf{H}_3 = [\mathbf{C}; \mathbf{O}],
\end{aligned}
\tag{4}
$$

where $[;]$ means the concatenation operation; $\mathbf{h}_1, \mathbf{h}_2$ and $\mathbf{h}_3$ are representations of the three hypernodes; $h_g$ denotes the score of the entire graph, meaning the extent that the current IP pertains to infeasibility or repetition; $\text{MLP}_3 \sim \text{MLP}_6$ refer to trainable scalar functions. We concretize $\text{MLP}_3 \sim \text{MLP}_5$ with two linear layers activated by ReLU function to compute attention weights, and concretize $\text{MLP}_6$ with two linear layers regularized by dropout in between. At last, we convert $h_g$ with Sigmoid function into a value in $[0, 1]$, which represents the probability of the current IP in the state being infeasible or repetitive. We keep the hidden dimension $d_r = 128$ throughout the GNN.

## 4.4 Training algorithm

We train the policy network by behavioral cloning [45], a specific imitation learning algorithm, which has been widely used in policy optimization to solve single-objective IPs [19, 21, 24, 37]. We sample the state-action data in ODAs by using the optimal policy in Definition 3 as the oracle, and then train the policy network in a supervised fashion.

The collection of states and the optimal actions is summarized by the pseudocode in Appendix A.5. We run an ODA and label states causing infeasible or repetitive IPs by $a_t = 1$, meaning the optimal policy is supposed to discard the corresponding local upper bounds $u_t$ without further solver calls. It is clear that the policy identified by the gathered state-action pairs is optimal for an MOIP instance according to Definition 3, since all nondominated points can be generated based on Proposition 2 and all infeasible or repetitive points are exactly avoided given $a_t = 1$. However, it would consume long time to collect all state-action pairs for an instance if it is complex or large-scale. In our training, we set a time limit $L$ to collect our data, since we emphasize on the bounded-time performance, which is also important in practical applications. With the collected dataset $D = \{(s_i, a_i)\}_{i=1}^I$, we train the policy network by minimizing the binary cross-entropy loss defined as below,

$$
\mathcal{L}(\theta) = -\mathbb{E}_D \, a_i \cdot \log \pi_\theta(a_i | s_i) + (1 - a_i) \cdot \log(1 - \pi_\theta(a_i | s_i)).
\tag{5}
$$

# 5 Experimental results

In the experiments, we deploy the proposed graph learning method on a state-of-the-art ODA [29], and evaluate it on two benchmark problems with varying numbers of objectives. We compare the learned algorithm with various baselines in literature, where multiple metrics are adopted. In addition, we assess the generalization of the learned reduction rules to MOIP instances which possess more objectives or larger sizes than the ones in training. Finally, we conduct the ablation study to verify the efficacy of the proposed two-stage GNN and the learned reduction rule.

## 5.1 Settings

We conduct experiments on two widely used benchmarks in the domain of MOIP, i.e., multi-objective knapsack problem (MOKP) and assignment problem (MOAP) [31], respectively.[4] Following the instance generation in [29], we uniformly sample the value and weight from $[1, 100]$ for each item in MOKP, and compute the capacity of the knapsack as half of the total weight. For MOAP, we randomly generate objective function coefficients (i.e. costs for each assignment) from $[1, 15]$.

**Instances & datasets.** For MOKP, we set the number of items to $100$ and generate $110$ instances with $3$ and $4$ objectives, respectively, with $100$ instances for training and $10$ for testing. We collect the training set by solving the $100$ instances with the ODA [29] and Gurobi solver (v9.5.1) [46], as stated in Section 4.4. While we solve 3-objective instances optimally (i.e. attain all Pareto-optimal points), we set $L = 1000s$ for 4-objective instances to collect the data so that we focus on training models for improving the bounded-time performance. We leave $1\%$ data in the training set for validation. In the same way, the testing sets are collected by solving the $10$ instances with $3$ and $4$ objectives, respectively. For MOAP, we set $50$ agents and $50$ tasks, respectively. We generate instances and collect the data in a similar way to MOKP except that we set $L = 1000s$ for all instances. In summary, we train, validate and test the algorithms on MOKP and MOAP instances with $3$ and $4$ objectives, respectively. We name these instance groups as MOKP(3-100), MOKP(4-100), MOAP(3-50) and MOAP(4-50), with the first value in parentheses being the number of objectives and the second identifying the problem size. We also generate $10$ instances for MOKP(4-150), MOKP(5-100), MOAP(4-75) and MOAP(5-50), respectively, to evaluate the generalization performance. Note that we set the number of objectives to the range $3\sim5$ as they are often used in ODAs [8, 30]. However, we also present more results with other numbers of objectives in Appendix A.6.

**Training & testing.** For training, we set the batch size to $64$ and initial learning rate to $0.001$. The learning rate decays by $0.2$ once the validation loss is not decreased in $8$ successive epochs. We employ early stopping to end the training when there is no improvement of validation performance in $16$ successive epochs. We always retrieve the GNN model with the lowest validation loss along epochs. We employ Adam optimizer [47] to minimize the loss and execute the training with one GeForce RTX 2080 Ti GPU. For testing, we plug the learned GNN model in the state-of-the-art ODA [29], and use the GNN assisted algorithm to solve testing instances in different groups. In specific, we discard a local upper bound to save the solving of the corresponding IP if the output value from our GNN is larger than $0.5$, so that the algorithm enters the next iteration by selecting another local upper bound, otherwise the algorithm will explore the IP by calling a solver (Gurobi in this paper) and update the search region as in the ODA. Except for GNN with GPU, the other algorithmic procedure is executed with one i9-10940X CPU@3.30GHz during testing.

## 5.2 Comparison study

We consider representative baselines of various paradigms for comparison, 1) three classic objective-space decomposition algorithms proposed by Kirlik and Sayın [31], Boland et al. [33], and Tamby and Vanderpooten [29] (the state-of-the-art, which is also used to deploy the proposed graph learning method), and we refer to them as ODA-K, ODA-B and ODA-T (it relies on Proposition 3 for infeasibility and solver for repetition), respectively; 2) the commonly used evolutionary algorithms including MOEAD, NSGAII, NSGAIII and UNSGAIII [48, 49, 50, 51]. We adapt and tune them for either of MOKP and MOAP [52, 53]; 3) the latest reinforcement learning model, i.e., PMOCO [10], which is specialized for MOCO problems with sequential structures. We adapt PMOCO to solve MOKP, which is trained on the same instances to ours, and other settings follow its original paper.

---

[4]http://home.ku.edu.tr/ moolibrary/

Table 1: Comparison with various baselines on MOKP and MOAP instances with 3 and 4 objectives.

| Methods | MOKP(3-100) | | | | MOKP(4-100) | | | | MOAP(3-50) | | | | MOAP(4-50) | | | |
|---|---|---|---|---|---|---|---|---|---|---|---|---|---|---|---|---|
| | Time(s) | Card. | HV | IGD | Time(s) | Card. | HV | IGD | Time(s) | Card. | HV | IGD | Time(s) | Card. | HV | IGD |
| ODA-T | 993.3 | **4169.8** | **0.55** | **0.000** | 1000.0 | 3691.4 | 0.75 | 0.012 | 1000.0 | 2532.9 | 0.81 | 0.009 | 1000.0 | 2663.7 | 0.52 | 0.017 |
| ODA-B | 1401.2 | **4169.8** | **0.55** | **0.000** | 1000.0 | 3188.4 | **0.76** | 0.007 | 1000.0 | 1736.5 | 0.86 | 0.005 | 1000.0 | 1576.5 | 0.66 | 0.018 |
| ODA-K | 928.4 | **4169.8** | **0.55** | **0.000** | 1000.0 | 3091.8 | 0.61 | 0.055 | 1000.0 | 2952.2 | 0.58 | 0.022 | 1000.0 | 2443.0 | 0.24 | 0.049 |
| MOEAD | 2000.0 | 1234.2 | 0.52 | 0.026 | 2000.0 | 1224.3 | 0.73 | 0.060 | 2000.0 | 984.1 | 0.76 | 0.008 | 2000.0 | 2282.7 | 0.51 | 0.017 |
| NSGAII | 2000.0 | 578.3 | 0.41 | 0.007 | 2000.0 | 998.8 | 0.68 | 0.021 | 2000.0 | 150.6 | 0.52 | 0.020 | 2000.0 | 893.6 | 0.29 | 0.029 |
| NSGAIII | 2000.0 | 390.0 | 0.53 | 0.005 | 2000.0 | 188.2 | 0.69 | 0.022 | 2000.0 | 482.3 | 0.69 | 0.017 | 2000.0 | 294.2 | 0.50 | 0.029 |
| UNSGAIII | 2000.0 | 545.7 | 0.41 | 0.007 | 2000.0 | 262.7 | 0.68 | 0.026 | 2000.0 | 401.9 | 0.69 | 0.018 | 2000.0 | 333.5 | 0.48 | 0.031 |
| PMOCO | 1000.0 | 228.4 | 0.53 | 0.008 | 1000.0 | 632.0 | 0.75 | 0.014 | - | - | - | - | - | - | - | - |
| Ours | **486.9** | 3379.7 | 0.55 | 0.000 | 1000.0 | **4000.7** | 0.76 | **0.006** | 1000.0 | **3564.4** | **0.87** | **0.004** | 1000.0 | **3771.3** | **0.67** | **0.011** |

Table 2: Generalization to larger sizes and 5 objectives.

| Methods | MOKP(4-150) | | | | MOKP(5-100) | | | | MOAP(4-75) | | | | MOAP(5-50) | | | |
|---|---|---|---|---|---|---|---|---|---|---|---|---|---|---|---|---|
| | Time(s) | Card. | HV | IGD | Time(s) | Card. | HV | IGD | Time(s) | Card. | HV | IGD | Time(s) | Card. | HV | IGD |
| ODA-T | 1000.0 | 3649.7 | 0.80 | 0.017 | 1000.0 | 2884.0 | 0.66 | 0.017 | 1000.0 | 1719.7 | 0.77 | 0.017 | 1000.0 | 2511.8 | 0.81 | 0.007 |
| ODA-B | 1000.0 | 2306.9 | 0.80 | 0.027 | 1000.0 | 2506.5 | 0.63 | 0.020 | 1000.0 | 899.1 | 0.73 | 0.037 | 1000.0 | 1312.9 | 0.75 | 0.028 |
| ODA-K | 1000.0 | 2588.0 | 0.53 | 0.109 | 1000.0 | 2654.2 | 0.45 | 0.110 | 1000.0 | 1640.3 | 0.24 | 0.064 | 1000.0 | 2236.9 | 0.36 | 0.047 |
| MOEAD | 2000.0 | 3834.1 | 0.75 | 0.089 | 2000.0 | *4200.9* | 0.60 | 0.084 | 2000.0 | *2383.6* | 0.45 | 0.037 | 2000.0 | *3221.5* | 0.59 | 0.026 |
| NSGAII | 2000.0 | 998.8 | 0.70 | 0.057 | 2000.0 | 1000.0 | 0.59 | 0.041 | 2000.0 | 930.7 | 0.24 | 0.062 | 2000.0 | 1105.7 | 0.34 | 0.043 |
| NSGAIII | 2000.0 | 185.6 | 0.72 | 0.058 | 2000.0 | 139.8 | 0.55 | 0.053 | 2000.0 | 209.9 | 0.46 | 0.062 | 2000.0 | 239.3 | 0.46 | 0.035 |
| UNSGAIII | 2000.0 | 264.5 | 0.71 | 0.069 | 2000.0 | 157.8 | 0.56 | 0.060 | 2000.0 | 233.2 | 0.40 | 0.056 | 2000.0 | 225.6 | 0.49 | 0.035 |
| PMOCO | 1000.0 | 732.0 | 0.81 | 0.023 | 1000.0 | 1475.5 | 0.66 | 0.020 | - | - | - | - | - | - | - | - |
| Ours | 1000.0 | **4032.6** | **0.82** | **0.013** | 1000.0 | **3634.4** | **0.66** | **0.015** | 1000.0 | **2137.4** | **0.79** | **0.017** | 1000.0 | **3089.2** | **0.82** | **0.006** |

We apply all methods on testing instances of MOKP(3-100), MOKP(4-100), MOAP(3-50) and MOKP(4-50), respectively. For MOKP(3-100), our method and three ODAs go through all local upper bounds since the instances are easy to be solved in relatively short time. For other instances, we set the time limit to 1000s for our method and ODAs. We set the same runtime for PMOCO across all instance groups and allow longer runtime (2000s) for evolutionary algorithms. To comprehensively assess the bounded-time performance, we adopt the following metrics, 1) Card.: cardinality of the returned set of nondominated points (for ODAs and ours) or feasible points (for others) [30];[5] 2) HV: hypervolume that reflects the convergence and spread of the approximated Pareto front [54]; 3) IGD: inverted generational distance that gauges the similarity between a set of images and the Pareto front [55]. We present details of the calculation for HV and IGD in Appendix A.7. For each metric, we record the average value across testing instances in each group and gather the results in Table 1.

For MOKP(3-100), we observe that the three classic ODAs attain all nondominated points and outperform other baselines in terms of the three metrics. However, our method is relatively fast to derive most nondominated points with less than half runtime. The nondominated points are very close to the Pareto front with almost the same HV and IGD as those in ODAs. In all approximate solutions, our method significantly surpasses evolutionary algorithms and PMOCO on all metrics. The advantage may stem from the facts that, 1) our method based on ODA-T adopts the highly-optimized IP solver (Gurobi) to fast compute nondominated points in scalarized problems, while solutions found in other heuristics have no guarantee to be nondominated; 2) the learned reduction rule further accelerates the search to find more nondominated points efficiently. We visualize the approximation (to Pareto front) by our method and heuristic baselines in Appendix A.8. Such advantage is also revealed on other instance groups, where our method consistently delivers higher HV and lower IGD than heuristic baselines. While ODA-T and ODA-B perform better than or comparable to the heuristics (in terms of HV and IGD) on MOKP(4-100), MOAP(3-50) and MOAP(4-50), our method generally outperforms the three ODAs on all metrics. On the other hand, although ODA-T has gained more nondominated points than ODA-B, ODA-K, our method with the learned reduction rule finds further more nondominated points than ODA-T in the same runtime, as shown in Card. columns. It

---

[5]While ODAs (thus also ours) ensure the optimal solutions to IPs are in the Pareto front according to Proposition 2, the other baselines belong to heuristic algorithms without any guarantee of solution quality.

Table 3: Ablation study on GNNs.

| Models | MOKP | | MOAP | | MOKP | | MOAP | |
|---|---|---|---|---|---|---|---|---|
| | 3-100 | 4-100 | 3-50 | 4-50 | 4-150 | 5-100 | 4-75 | 5-50 |
| GCN | 68.4 | 67.6 | 72.7 | 91.2 | **70.4** | 58.8 | 94.5 | 90.6 |
| GraphSage | 63.0 | 70.8 | 69.6 | 91.2 | 59.5 | 59.9 | 94.8 | **90.7** |
| GIN | 79.7 | 82.0 | 79.6 | 91.2 | 59.5 | 58.8 | 93.4 | 90.6 |
| 2-stage GIN | 80.7 | **82.5** | 80.8 | **91.3** | 59.5 | 58.8 | **95.2** | 90.6 |
| Ours | **82.8** | 80.9 | **82.8** | **91.3** | 68.6 | **73.9** | 95.0 | **90.7** |

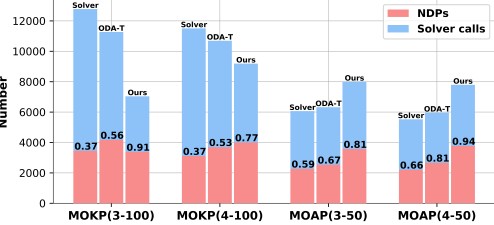

Figure 3: Ablation study on reduction rules.

indicates that our method is more effective in curtailing the solving of futile IPs, which could save runtime to explore more fruitful IPs. Meanwhile, the nondominated points found by our method show desirable convergence and spread, which are identified by higher HV and lower IGD values.

## 5.3 Generalization study

The generalization to instances larger than the ones in training is critical, as it may determine whether the learned algorithm could be used to solve a wider range of instances. Meanwhile, the generalization to more objectives is also valuable for MOIPs, where a decision maker may freely consider additional objectives through the learned algorithm. To this end, we apply the learned ODAs trained on MOKP(4-100) and MOAP(4-50) to solve instances with larger scales and more (i.e. 5) objectives, respectively. We also adapt evolutionary algorithms and PMOCO to solve these instances. Specially, we directly train PMOCO on 5-objective instances following [10], since it cannot generalize across objectives. All results are displayed in Table 2. It is manifested that our method attains the best values on all metrics for different problems. While MOEAD achieves significantly more feasible points and higher HV than other evolutionary algorithms and ODA-K, it performs inferior to PMOCO, ODA-T and ODA-B. Pertaining to ODAs, we observe that ODA-T generally outperforms ODA-B and ODA-K on all metrics and is comparable to PMOCO. However, our method with the learned reduction rule further improves ODA-T by circumventing futile IP solving, which also outstrips PMOCO. With the highest HV and lowest IGD, our method yields a better approximation to the Pareto front, implying that the learned reduction rule can generalize well to harder instances.

## 5.4 Ablation study

To evaluate the efficacy of our two-stage GNN, we compare it with the classic bipartite GCN [21], the latest bipartite GIN and GraphSage [39], all of which were used to solve IPs in their original papers. We arrange results in Table 3 where *test* and *generalization* accuracy are placed in the left and right half, respectively. We observe that our model generally transcends other GNNs for testing, except for MOKP(4-100) with a comparable accuracy to GIN. Meanwhile, our model delivers the highest generalization accuracy on most instance groups, except for MOKP(4-150), where it performs slightly inferior to GCN. We also verify the efficacy of our aggregation scheme (in the second stage) by equipping it to GIN given the better test performance than other baselines, which results in the two-stage GIN. As shown, the two-stage GIN improved the accuracy over GIN on $5/8$ instance groups, implying our scheme is effective to gain a better graph representation. However, our GNN still achieves superior overall performance to this enhanced GIN, suggesting that our attention in the first stage attains better node embeddings that benefit the subsequent graph representation learning.

Furthermore, we compare our method with ODA-T regarding the usage rate of the IP solver (Gurobi), to show the efficacy of the learned reduction rules. A vanilla version of ODA-T is also tested where we purely use solver to detect infeasibility and repetition (note: different from its vanilla version, ODA-T relies on Proposition 3 for infeasibility and solver for repetition). We plot the average numbers of nondominated points, solver calls and ratio in Figure 3. As shown, our method attains the most nondominated points on $3/4$ instance groups except for MOKP(3-100) which is easy for ODA-T to solve optimally. While the ratio of ODA-T is higher than its vanilla version, the learned reduction rule further enhances ODA-T with (at least) $13\%$ higher ratio on different groups. It means the reduction rule assisted by our GNN is more effective than the one in ODA-T.

# 6 Conclusions and future work

This paper presents a graph neural network (GNN) based method to learn the reduction rule in an ODA, with the aim of improving its bounded-time performance in solving MOIPs. We train a novel two-stage GNN with imitation learning to discard local upper bounds that induce infeasibility and repetition of nondominated points. We implement our method with a state-of-the-art ODA and empirically show that it can significantly enhance the ODA and generalize to larger problems or more objectives. We also verify the superiority of the proposed GNN to existing ones for integer programs, in terms of better test and generalization accuracy. The limitation of our method may appear when the IP solver cannot fast solve intractable scalarized problems. A solution could be compromising the optimality in Proposition 1 and replace IP solver with heuristics, so that they could still be guided by the schemes in ODAs and the learned reduction rule. We leave this interesting direction as future work. Also, we plan to deploy our method to more ODAs.

## Acknowledgments and Disclosure of Funding

This work was supported in part by the A*STAR Cyber-Physical Production System (CPPS) - Towards Contextual and Intelligent Response Research Program, under the RIE2020 IAF-PP grant A19C1a0018, and Model Factory@SIMTech; in part by the A*STAR HTPO seed grant C211118016, and the A*STAR Career Development Fund under grant C222812027; in part by the National Natural Science Foundation of China under grant 62102228, and the Shandong Provincial Natural Science Foundation under grant ZR2021QF063.

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
