# Graph Learning Assisted Multi-objective Integer Programming (Appendix)

## A.1    Search region update

According to Definition 2 in the main paper, the search region is described by a set of (maximal) local upper bounds. Hence, we update the search region by efficiently updating the local upper bounds when a new nondominated point is attained based on Eq. (2). We summarize the update procedures in Algorithm 1 as below. It is originally proposed in [26] and used by the ODA in [27], on which we deploy the graph learning method.

---
**Algorithm 1:** Update of local upper bounds

---
1: **Input:** A set of points $N$, the corresponding local upper bounds $U(N)$, a new nondominated point $y^*$
   (Note: $U(N)$ is initialized as $U(\emptyset) = \{(M, \ldots, M)\}$ as introduced in the main paper)
2:  $N' = N \cup y^*; U(N') = U(N);$
3:  **for** $u \in U(N)$ **do**
4:      ## if $y^*$ dominates $u$, $u$ is deleted and new bounds are added;
       or, $y^*$ could be a defining point of $u$. ##
5:      **if** $y^* < u$ **then**
6:          $U(N') = U(N')\backslash\{u\};$
7:          **for** $i \in \{1, \ldots, p\}$ **do**
8:              $u^i = (u_1, \ldots, u_{i-1}, y_i^*, u_{i+1}, \ldots, u_p);$
9:              $N_i(u^i) = \{y^*\}$
10:             **for** $j \in \{1, \ldots, p\}\backslash\{i\}$ **do**
11:                 $N_j(u^i) = \{y | y \in N_j(u), y_i < y_i^*\}$
12:             **if** $N_j(u^i) \neq \emptyset, \forall\{j | j \in \{1, \ldots, p\}\backslash\{i\}, u_j^i \neq M\}$ **then**
13:                 $U(N') = U(N') \cup \{u^i\}$
14:     **else**
15:         **for** $i \in \{1, \ldots, p\}$ **do**
16:             **if** $y_i^* = u_i$ and $y_{-i}^* < u_{-i}$ **then**
17:                 $N_i(u) = N_i(u) \cup \{y^*\}$
18: **Return:** updated local upper bounds $U(N')$

---

In the above algorithm, $N_i(u)$ denotes the defining points for the $i$-th dimension of the local upper bound $u$. It satisfies, 1) $N_i(u) \subset N$, and 2) $\forall y \in N_i(u)$, $y_k = u_k$ and $y_{-k} < u_{-k}$, where $y_{-k}$ refers to a reduction of $y$ with its $k$-th component eliminated. According to [26, 27], the set of local upper bounds is maximal if and only if every bound in the set has at least one defining point on the dimensions $\{i | i \in \{1, \ldots, p\}, u_i \neq M\}$. Therefore, Algorithm 1 adds new local upper bounds by ensuring they have defining points for bounded dimensions (line 7 $\sim$ 13). When $y_i^* = u_i$ and $y_{-i}^* < u_{-i}$, $y^*$ is exactly a defining point of $u$ and added to $N_i(u)$ (line 15 $\sim$ 17). We refer the readers to [26, 27] for more details.

## A.2    Scalarizations

ODAs normally rely on scalarization methods to convert MOIP into a series of single-objective IPs. A variety of scalarization methods have been proposed for different ODAs. Given a MOIP as in Eq. (1) in the main paper, we introduce four commonly used methods including the one (i.e. Lexicographic optimization) used in our approach. More scalarization methods can be found in [6, 8].

36th Conference on Neural Information Processing Systems (NeurIPS 2022).

**Weighted sum method.** It minimizes a weighted sum of objectives as follows: $\min_x \{\sum \mu_i f_i(x) | \mu \in \mathbb{R}^p; x \in \mathcal{X}\}$. The optimal solution to such scalarization is an (weakly) efficient solution to the MOIP when $\mu (\geq) > 0$. While this method is simple and widely used, it is not appropriate for nonconvex objective spaces since some nondominated points cannot be represented by the scalarized objective.

**$\epsilon$-constraint method.** It optimizes one objective with constraints to other objective values as follows: $\min_x \{f_k(x) | f_i(x) \leqslant \epsilon_i, i \neq k, \epsilon \in \mathbb{R}^p; x \in \mathcal{X}\}$. Its optimal solution is weakly efficient for MOIP, and many variants have been proposed to derive efficient solutions. Classic $\epsilon$-constraint methods enumerate $\epsilon$ to compute all nondominated points and require considerable iterations of IP solving when the number of objectives rises. ODAs can be used as a paradigm to improve the classic methods since they efficiently exclude the dominated region by newly derived points in the objective space.

**Weighted Tchebycheff method.** It minimizes the maximal distances between objectives and the ideal point $y^I \in \mathbb{R}^p$ as follows: $\min_x \{\max_{1 \leqslant i \leqslant p} \{\mu_i (f_i(x) - y_i^I)\} | x \in \mathcal{X}\}$, where $\mu \geq 0$ captures the preference over different objectives. The optimal solution to the above formulation is weakly efficient to MOIP, and variants have been developed to compute efficient solutions, e.g., lexicographic and augmented weighted Tchebycheff methods.

**Lexicographic optimization method.** It is used in the ODA [27], which is defined as follows: $\text{lexmin}\{\{f_k(x), \sum_{i=1, i \neq k}^{p} f_i(x)\} | f_i(x) \leqslant u_i, i \neq k, u \in \mathbb{R}^p; x \in \mathcal{X}\}$. The $k$-th objective is optimized in priority and the sum of the other objectives is optimized without impairing the optimal value of the $k$-th objective. Similar to the $\epsilon$-constraint method, $u$ is used to restrict the objective values, i.e., the search region for the above IP in the objective space.

## A.3   Proofs

Here we rewrite Proposition 1 and 2 and present proofs for them.

**Proposition 1.** Given $u \in U(N)$ and a strictly monotone function $\mathcal{F} : \mathcal{Y} \to \mathbb{R}$, the optimal solution to the following scalarized problem (IP) will derive a nondominated point in $\mathcal{Y}_{nd}$,

$$\Gamma(u) := \min\{\mathcal{F}(y) = \mathcal{F}(y_1, \ldots, y_p) | y \in \mathcal{Y}; y < u\}. \tag{A.1}$$

**Proof.** We assume the optimal solution $y^*$ to $\Gamma(u)$ is not a nondominated point, i.e., $y^* \notin \mathcal{Y}_{nd}$. Therefore, there exists a point $y' \in \mathcal{Y}_{nd}$ satisfying $y' \leq y^* < u$. According to the strictly monotone property of $\mathcal{F}$, we have $\mathcal{F}(y') < \mathcal{F}(y^*)$ which contradicts with the optimality of $y^*$ in $\Gamma(u)$.

**Proposition 2.** ODAs generate the whole Pareto front with finite iterations.

**Proof.** Given the bounded MOIP defined in Eq. (1) in the main paper, the number of nondominated points is finite, i.e., $|\mathcal{Y}_{nd}| < \infty$. The ODAs iteratively solve IPs with the form in Eq. (A.1), and each arrives at the following possible results: 1) a new nondominated point, 2) a repetitive nondominated point, and 3) infeasibility. According to Definition 2, the remaining nondominated points are located in the search region delimited by the local upper bounds. Thus solving IPs (constrained by the bounds) optimally in iterations will attain all nondominated points, according to Proposition 2. Meanwhile, the local upper bounds inducing infeasible IPs and repetitive points are ignored by ODAs, which means the regions without nondominated points are excluded. Hence, the number of iterations cannot be infinite within the bounded and discrete objective space.

## A.4   Featurization

Given the graph encoding in the main paper, we can readily represent different scalarized problems by different featurizations on the graph. Here, we realize the featurization of the Lexicographic scalarizations for the ODA in [27], since we use it to deploy our graph learning method. According to the original work in [27], the Lexicographic scalarization is defined as follows,

$$\text{lexmin} \quad \left\{ f_k(x), \sum_{i=1, i \neq k}^{p} f_i(x) \right\} \tag{A.2}$$
$$\text{s.t.} \quad f(x) \in \mathcal{Y}, \ f_i(x) \leqslant u_i, \forall i \in \{1, \ldots, p\}, i \neq k$$

Table A.1: Featurization of the Lexicographic scalarization on the graph.

| Features | Description | Digit |
|---|---|---|
| Objective features ($\mathbf{O}$) | The ideal point $y_i^I = \min_{x \in \mathcal{X}} f_i(x)$. | 1 |
| | The selected local upper bound $u$. | 1 |
| | The defining point as described in Section A.1. | 1 |
| | The average, maximum and minimum of current local upper bounds. | 3 |
| | One-hot vector to identify which objective is in priority. | 1 |
| Constraint features ($\mathbf{C}$) | The right-hand side of constraints in Eq. (A.2). | 1 |
| Variable features ($\mathbf{V}$) | The average and standard deviation of history solutions. | 2 |
| | Coefficients of variables for the objective in priority. | 1 |
| Edge features ($\mathbf{E}_{ov}$) | Coefficients of variables for all objectives. | 1 |
| Edge features ($\mathbf{E}_{cv}$) | Coefficients in the incidence matrix between variables and constraints. | 1 |

where $u = (u_1, \ldots, u_p)$ is the selected local upper bound in an iteration of ODA. The logic behind the above Lexicographic scalarization is that the $k$-th objective is optimized in priority, so that the sum of the remaining objectives are optimized without impairing the optimized value of the $k$-th objective. The ODA in [27] employs IP solver to solve the above scalarized problems (IPs) efficiently.

Our work centers on learning representations of scalarized problems in the above form to predict whether the corresponding IPs 1) are infeasible, or 2) lead to repetition of nondominated points, with the proposed two-stage GNN. Based on the prediction, we choose to solve the scalarized problem, or discard the selected local upper bound and enter the next iteration. To effectively capture the above scalarized problem, we leverage a set of features as described in Table A.1.

## A.5 Pseudocode for dataset collection

---
**Algorithm 2:** Data collection for a MOIP instance with an ODA
---
1: **Input:** MOIP instance $z$, local upper bounds $U = U(\emptyset)$, nondominated points $N = \{\}$, time limit $L$, step $t = 0$, dataset $D_z = \{\}$
2: **while** runtime $< L$ **do**
3:      Select a local upper bound $u_t$ from $U$;
4:      Record the current state $s_t$ by collecting the features in Section A.4;
5:      Solve the IP defined by $u_t$ with the solver;
6:      **if** IP is infeasible or returned point $y$ is already in $N$ **then**
7:          $D_z = D_z \cup (s_t, 1)$; Discard $u_t$ from $U$;
8:      **else**
9:          $D_z = D_z \cup (s_t, 0)$; Update $U$ by ODA itself as described in Section A.1; $N = N \cup y$
10:      **if** U={} **then**
11:          **Return:** $D_z$
12:      $t = t + 1$;
13: **Return:** $D_z$
---

According to the optimal policy in Definition 3 in the main paper, we collect states and the optimal actions for each instance following the procedure in Algorithm 2. We run an ODA and label states which induce infeasible IPs or repetitive nondominated points, by $a_t = 1$. The proposed graph learning approach aims to approximate such an optimal policy by discarding the local upper bounds $u_t$ in futile IPs. We execute Algorithm 2 for every training instance in each group and collect the training set $D = \bigcup_{z=1}^{Z} D_z = \{(s_i, a_i)\}_{i=1}^{I}$. Except for the component we plug in to extract features, the other components remain the same as in the original ODA, such as the selection of the bounds at line 3 and the update of the search region at line 9.

## A.6 Additional results

We present more results on instances with 6 and 7 objectives. Following the instance generation in the main paper, we generate 110 instances of MOKP(6-100) and MOAP(6-40), respectively,

Table A.2: Comparison with baselines on MOKP and MOAP instances with 6 and 7 objectives.

| Methods | MOKP(6-100) | | | | MOKP(7-100) | | | | MOAP(6-40) | | | | MOAP(7-50) | | | |
|---|---|---|---|---|---|---|---|---|---|---|---|---|---|---|---|---|
| | Time(s) | Card. | HV | IGD | Time(s) | Card. | HV | IGD | Time(s) | Card. | HV | IGD | Time(s) | Card. | HV | IGD |
| ODA-T | 1000.0 | 2554.6 | 0.53 | 0.059 | 1000.0 | 1946.3 | 0.45 | 0.080 | 1000.0 | 1976.3 | 0.76 | 0.016 | 1000.0 | 951.5 | 0.80 | 0.024 |
| ODA-B | 1000.0 | 1684.2 | 0.53 | 0.064 | 1000.0 | 1099.6 | 0.49 | 0.081 | 1000.0 | 1509.8 | 0.73 | 0.019 | 1000.0 | 980.9 | 0.83 | 0.023 |
| ODA-K | 1000.0 | 2437.1 | 0.40 | 0.075 | 1000.0 | 2025.9 | 0.31 | 0.104 | 1000.0 | 1966.5 | 0.35 | 0.037 | 1000.0 | 1228.5 | 0.45 | 0.051 |
| MOEAD | 2000.0 | *3612.5* | 0.50 | 0.061 | 2000.0 | *4345.1* | 0.46 | 0.072 | 2000.0 | *2958.9* | 0.63 | 0.021 | 2000.0 | *3235.4* | 0.67 | 0.033 |
| NSGAII | 2000.0 | 1000.0 | 0.47 | 0.075 | 2000.0 | 1000.0 | 0.40 | 0.065 | 2000.0 | 1793.5 | 0.35 | 0.035 | 2000.0 | 2000.0 | 0.43 | 0.052 |
| NSGAIII | 2000.0 | 140.2 | 0.44 | 0.084 | 2000.0 | 150.3 | 0.35 | 0.090 | 2000.0 | 241.1 | 0.44 | 0.029 | 2000.0 | 243.1 | 0.49 | 0.045 |
| UNSGAIII | 2000.0 | 149.7 | 0.45 | 0.085 | 2000.0 | 121.8 | 0.38 | 0.099 | 2000.0 | 235.9 | 0.45 | 0.028 | 2000.0 | 237.7 | 0.52 | 0.044 |
| PMOCO | 1000.0 | 2464.7 | 0.53 | 0.061 | 1000.0 | 2688.5 | **0.51** | **0.064** | - | - | - | - | - | - | - | - |
| Ours | 1000.0 | 2996.4 | **0.54** | **0.057** | 1000.0 | 2392.8 | 0.51 | 0.065 | 1000.0 | 2261.3 | **0.80** | **0.009** | 1000.0 | 1358.2 | **0.87** | **0.012** |

with 100 instances for the collection of training set (in which $1\%$ is used for validation) and 10 instances for testing. We also generate 10 instances of MOKP(7-100) and MOAP(7-50) to evaluate the generalization of the models trained with MOKP(6-100) and MOAP(6-40), respectively. We train PMOCO with instances of MOKP(6-100) and MOKP(7-100) since it cannot generalize across different numbers of objectives. The results are displayed in Table A.2.

We observe that our method achieves the highest HV and the lowest IGD values for MOKP(6-100), MOAP(6-40) and MOAP(7-50). While it performs slightly inferior to PMOCO for MOKP(7-100), i.e., 0.508/0.512 w.r.t HV and 0.065/0.064 w.r.t IGD, we note that our model is directly applied to instances with more objectives and PMOCO has no such generalization capability. Particularly, without the need of extra efforts to train the model with new data, our method attains comparable results to PMOCO. This advantage allows our method to be more preferable in practice where a decision maker can add objectives freely and get the results quickly. Furthermore, our method significantly improves ODA-T with more nondominated points found in the same runtime, and the cardinality is also larger than those of ODA-B, ODA-K and most heuristic baselines. While MOEAD achieves the largest cardinality in each instance group and mostly attains the best performance among evolutionary algorithms, it is still inferior to ODA-T, ODA-B and our method. This stems from the fact that the (feasible) solutions returned by MOEAD are of poor quality and ODAs (including ours) guarantee that the solutions are exactly nondominated. In summary, our method not only achieves the best performance for instances that are similar to the ones in training, but also generalizes fairly well to instances with more objectives and larger sizes.

## A.7 Computation of HV and IGD

**Hypervolume.** The hypervolume for a set of feasible points $F$ is defined as the volume of a subspace in the objective space, which is weakly dominated by $F$ and bounded by a reference point $r^*$, as follows,

$$\text{HV}(F) = \lambda^p(\{y \in \mathbb{R}^p | \exists y' \in F, y' \leq y \leq r^*\}), \tag{A.3}$$

where $\lambda^p$ denotes the Lebesgue measure on the $p$-dimensional space which calculates the volume for a $p$-dimensional subspace. The hypervolume is a widely used metric to assess the performance of approximated solutions. Since the range of objective values varies across different problems and instances, we employ the normalized hypervolume (NHV) following [10], which is defined as $\text{NHV}(F) = \text{HV}(F) / \prod_{i=1}^p r_i^*$. We compute $r^*$ for an instance as element-wise maximums of all points from different methods, so that NHVs of solutions from each method are derived for this instance. We then compute the average NHV across the instances used for testing.

**Inverted generational distance.** Given a set of feasible points $F$ and the reference set $Y$, the inverted generational distance is defined as follows,

$$\text{IGD}_\sigma(Y, F) = \frac{1}{|Y|} \Big( \sum_{y \in Y} \min_{f \in F} d(y, f)^\sigma \Big)^{1/\sigma}, \tag{A.4}$$

where $d(y, f) = (\sum_{i=1}^p (y_i - f_i)^2)^{1/2}$ represents Euclidean distance and $\sigma$ is set to 1. The reference set is set as the Pareto front if all nondominated points are known. However, it is often not practical

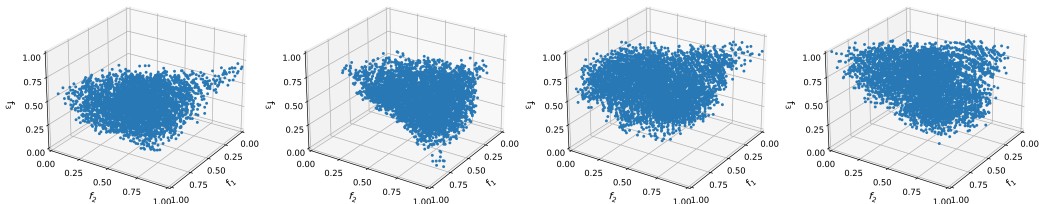

Figure A.1: The exact Pareto front.

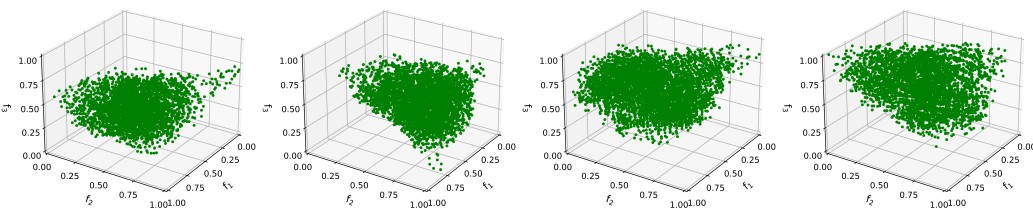

Figure A.2: Approximated Pareto front by our method.

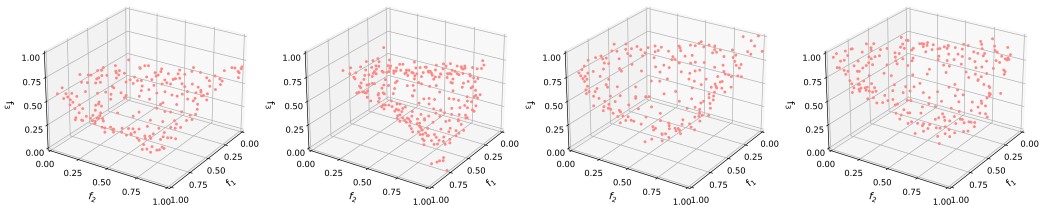

Figure A.3: Approximated Pareto front by PMOCO.

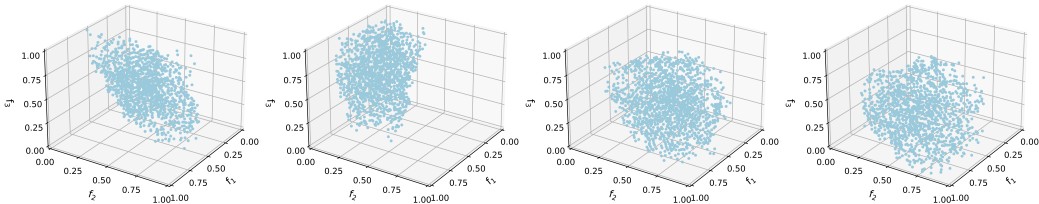

Figure A.4: Approximated Pareto front by MOEAD.

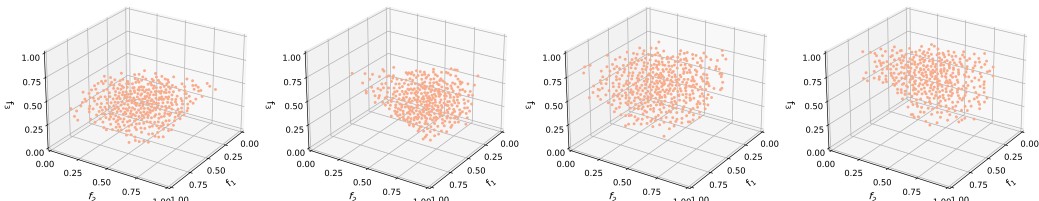

Figure A.5: Approximated Pareto front by NSGAIII.

for complex MOIPs in which the whole Pareto front is intractable to compute in prior. On the other hand, the reference set could also be a (good) approximated Pareto front for assessment. In this paper, we use the exact Pareto front for MOKP(3-100) to compute IGD since they are easy to be solved optimally. For the other instances, we gather all nondominated points from all methods as the good approximated Pareto front. This approximation is closer to the true Pareto front and thus better than the approximated solutions from each method. Moreover, we find that it is better than the one achieved by running a single method with long time, since different methods could generate different parts of nondominated points in Pareto front and we gather them together to gain better spread.

## A.8 Visualization

We visualize the solutions attained by different methods. We pick four representatives from all baselines in the main paper including ODA-T in ODAs, MOEAD, NSGAIII in evolutionary computation, and the recent PMOCO in deep reinforcement learning. Among them, ODA-T achieves the exact Pareto front. Regarding each baseline, we plot solutions to four testing instances for MOKP(3-100) by normalizing the objective values to $[0, 1]^{\mathbb{R}=3}$, which are displayed in Figure A.1∼ A.5. It is clear that the solutions achieved by our method is almost the same to the exact Pareto front, with good convergence and spread. In comparison, MOEAD and NSGAIII exhibit clear biases towards only parts of the Pareto front. While PMOCO scatters its points dispersedly with a similar shape to the Pareto front, the points are too sparse to approximate the details. Furthermore, our method guarantees the derived points are exactly nondominated (i.e. in Pareto front) and thus gains a substantially better approximation. Last but not least, our method consumes significantly less runtime than other methods (i.e. less than half) as stated in the main paper, which suggests a favorable trade-off to fast achieve an adequately good approximated Pareto front.