# OpenReview forum: "Graph Learning Assisted Multi-Objective Integer Programming"
_NeurIPS.cc/2022/Conference — NeurIPS 2022 Accept_

### Official Review · Reviewer_LVBy · 2022-07-09

**Rating:** 7
**Confidence:** 4
**Soundness:** 3 good
**Presentation:** 3 good
**Contribution:** 3 good

**Summary:**

This work proposes a learning-based approach to enhance objective-space decomposition (ODA) for solving multi-objective integer programs (MOIP). Specifically, it builds a novel two-stage GNN as a parameterized policy to learn the reduction rule for ODA by imitation learning. The proposed GNN model takes the instance information (decision variables, constraints, and objectives) as input, and then adaptively discards local upper bounds (to update the search region) without calling an IP solver in ODA. In this way, it can fast detect and eliminate the infeasible and repetitive IPs, and hence reduce redundant computation. Experimental results show that the proposed method can significantly reduce the runtime for a state-of-the-art ODA, and can be generalized well to solve problems with larger sizes or more objectives.

**Questions:**

- In my understanding, the proposed method is only trained with the data generated by ODA-T. Is it possible to train it with data from ODA-T/B/K at the same time? Will this approach lead to more robust performance?

- The proposed method can outperform all ODAs with a bounded time, but it is also interesting to know its unbounded performance. Given enough run time, how many valid IPs could be wrongly eliminated by the proposed method?

**Limitations:**

The limitations of this work are 1) the requirement of ODAs and a well-designed IP solver; 2) it loses the guarantee for finding the whole Pareto front. They have been properly discussed in the paper (see remark at the end of Section 4.1 and Conclusion).

I do not see any potential negative societal impact of this work.

**Strengths And Weaknesses:**

**Strengths:**

+ This work is generally well-written and easy to follow.

+ MOIP problems can be found in many real-world applications, while they are usually very hard to be solved optimally. Learning-assisted method for solving MOIP is an important research topic, and this work makes a timely contribution in this direction.

+ In my understanding, it is the first work on learning-assisted MOIP. The proposed GNN-based policy to enhance ODA is sound and novel, which can inspire more future work.

+ It is promising that the learned policy can generalize well to solve problem instances with different sizes and number of objectives, which could be important for real-world applications.

**Weaknesses:**

**1. Imitation Learning:** The proposed method needs to be trained by behavioral cloning, which means 1) it requires a carefully well-designed algorithm (e.g., ODA-T/B/K) to generate the supervised data set. 2) More importantly, the data generated by ODA with a time limit L is indeed not a perfect teacher for behavioral cloning. Since all the ODAs are not designed and optimized for bounded time performance, their behavior (the state-action pairs) for the first L time is not the optimal policy under time limit L.

Since the generic algorithm procedure of ODA can be encoded as a MDP, is it possible to use reinforcement learning to train the model? Would the RL-based approach find a better policy for bounded time performance that is aware of the time limit T？

**2. Performance with Different Time Limits:** It seems that the proposed method is both trained and tested with a single fixed time limit T = 1000s for all problems. However, in practice, the applications could have very different run time limits. Will the proposed method generalize well to different time limits (such as 1/10/100/2000/5000s)?

**3. Comparison with PMOCO:** PMOCO is the only learning-based approach in the comparison, but it has a very small cardinality (feasible points) for most problems. Its approximated Pareto front is also relatively sparse (which means a small set of solutions) in Figure A.3. This result is a bit counter-intuitive.

In my understanding, PMOCO is a construction-based neural combinatorial optimization algorithm, of which one important advantage is the very fast run time. The PMOCO paper [1] reports it can generate 101 solutions for 100 MOKP(2-100) instances (hence 10,100 solutions in total) in only 15s, and 10,011 solutions for 100 MOTSP(3-100) instances (hence 1,001,100 solutions in total) in 33 minutes (~2000s). In this work, with a large time limit of 1,000s, I think POMO should be able to generate a dense set of solutions for each instance.

In addition, while a dataset with 100 instances could provide enough supervised ODA state-action pairs (with a time limit L = 1000s for each instance) for imitation learning, it is far from enough for PMOCO's RL-based training. Since PMOCO does not require any supervised data and the MOKP instances can be easily generated on the fly, is it more suitable to train PMOCO under the same wall-clock time with the proposed method?

[1] Pareto set learning for neural multi-objective combinatorial optimization. ICLR 2022.

---

> ### Author Response · Authors · 2022-08-02
> **Response to Reviewer LVBy (3/3)**
>
> **Q5) Regarding the potential of our method to train PMOCO.**
>
> It is an interesting idea to train PMOCO with our method, as mentioned by the reviewer. To realize it, we can firstly collect the solutions  which correspond to the nondominated points and the local upper bounds which define the IPs of the solutions,  by running ODAs to solve training instances with a time limit. All the collected solution-local upper bound pairs comprise a dataset to train PMOCO. During training, we input local upper bounds (which correspond to the preferences in original PMOCO [3]) as the perturbation on the decoder and use solutions as labels to train the neural network in a supervised manner, as did in [4]. During testing,  we need to  input local upper bounds sequentially to infer the solutions. To derive these bounds, the update rule of local upper bounds in Appendix A.1 can be used.
>
> Given the above potentially possible realization, it indicates that our method  could also be used to aid the policy training for specific MOIP problems. For example, we can pre-train PMOCO with our method to derive a good initialized policy for RL, so that we can mitigate the considerable time to explore from a randomly initialized policy.
>
> **Q6) Regarding the training with mixed data generated by multiple ODAs.**
>
> The idea to train our model with mixed data from ODA-T/B/K is quite interesting. However, we would like to note that different ODAs might be characterized by different scalarization techniques, which also means different featurizations for graph encoding. It may hinder the mixed use of samples from different ODAs. To alleviate the limitation, one can use ODAs with similar scalarization techniques at the same time, so that the policy can be trained with these similar problem domains. In doing so, the mixed dataset from good behaviors of different ODAs may aid the policy learning for a more robust performance, since each of ODAs might be superior on different groups of instances and the learned policy on the mixed dataset could aggregate the advantage from these ODAs.
>
> **Q7) Regarding the evaluation of unbounded performance.**
>
> Please note that the current work mainly centers on the evaluation of bounded-time performance of the proposed method, since it is more practical. Furthermore, considering our work as a very early attempt to solve MOIP with graph learning, it is critical to verify the basic abilities of the learned policy and the effectiveness of the proposed GNN. Hence we conduct extensive experiments for the comparison, generalization and ablation studies, which we find are mostly concerned in the learning-to-optimize community.
>
> On the other hand, some of our results in our paper already reflect the mentioned unbounded performance. For example, according to the results for MOKP(3-100) in Table 1, our method gains 3379.7 nondominated points on average, while 4169.8 nondominated points exist in the exact Pareto front. It means we have found 81.1% nondominated points, and wrongly deleted 18.9%. Despite the wrong deletion, our method costs very short time (i.e. merely half time compared to the exact method) to find most of nondominated points. Again, this advantage is more obvious on large and complex problems. Our method can gain significantly more nondominated points than either exact or heuristic baselines on MOKP(4-100), MOAP(3-50) and MOAP(4-50), with better HV and IGD values.
>
> Finally, we acknowledge that  it is also meaningful to comprehensively check the unbounded performance of the proposed method, e.g., the total wrongly discarded IPs, on different problem sizes and types. We will leave this evaluation as our future work.
>
> Reference:
>
> [1] Pascal Halffmann, Luca E Schäfer, Kerstin Dächert, Kathrin Klamroth, and Stefan Ruzika. Exact algorithms for multiobjective linear optimization problems with integer variables: A state of the art survey. Journal of Multi-Criteria Decision Analysis, 2022.
>
> [2] Fabio Pardo, Arash Tavakoli, Vitaly Levdik and Petar Kormushev. Time Limits in Reinforcement Learning. In Proceedings of the 35th International Conference on Machine Learning (ICML), 2018.
>
> [3] Xi Lin, Zhiyuan Yang, and Qingfu Zhang. Pareto set learning for neural multi-objective combinatorial optimization. In Proceedings of the 10th International Conference on Learning Representations (ICLR), 2022.
>
> [4] Oriol Vinyals, Meire Fortunato, Navdeep Jaitly. Pointer networks.In Proceedings of the 29th Conference on Neural Information Processing Systems (NeurIPS), 2015.

---

> > ### Comment · Reviewer_LVBy · 2022-08-05
> > **Thank you for the thorough response**
> >
> > Thank you for the thorough and to-the-point response. All of my concerns have been properly addressed, so I raise my score to 7.
> >
> > As fairly discussed by the author in the paper and in the response, the proposed method might have some limitations, such as 1) requiring a well-designed ODA and IP solver; 2) not taking the bounded time into consideration, etc. But I think the current work has already made a good contribution in proposing the very first graph learning-assisted MOIP solver, which could inspire more follow-up work on this important research direction. Therefore, I vote for accepting this paper.

---

> > > ### Author Response · Authors · 2022-08-05
> > > **Response to Reviewer LVBy**
> > >
> > > Thanks for your acknowledgement and support.

---

> ### Author Response · Authors · 2022-08-02
> **Response to Reviewer LVBy (2/3)**
>
> **Q3) Regarding the generalization to different time limits.**
>
> As a very early attempt to solve MOIP with graph learning, we focus more on its comparison with conventional baselines, and generalization across problem sizes and number of objectives (note: it has not been done in previous related works). We acknowledge that the generalization to time limits is meaningful but beyond the scope of the current work. However, we conduct a quick experiment to evaluate our trained policies (1000s) with two different time limits, i.e., 500s and 2000s. The other baselines are also evaluated with the same adjusted time limits, except the evolutionary algorithms (with 1000s and 3000s). Other experimental setting is the same as the one in Section 5.2, and the results are shown in the following table. We observe that our method gains the largest cardinalities on all instance groups. Meanwhile, it achieves the highest HV and lowest IGD on MOAP(4-50) with 500s, as well as MOKP(4-100) and MOAP(4-50) with 2000s. It indicates that our method generalizes well to different time limits.
>
> |          |         | MOKP(4-100)-500s |      |       |         | MOKP(4-100)-2000s |      |       |         | MOAP(4-50)-500s |      |       |         | MOAP(4-50)-2000s |      |       |
> |----------|---------|------------------|------|-------|---------|-------------------|------|-------|---------|------------------|------|-------|---------|-------------------|------|-------|
> | Methods  | Time(s) | Card.            | HV   | IGD   | Time(s) | Card.             | HV   | IGD   | Time(s) | Card.            | HV   | IGD   | Time(s) | Card.             | HV   | IGD   |
> | ODA-T    | 500s    | 3203.9           | 0.75 | 0.010 | 2000s   | 5818.6            | 0.76 | 0.006 | 500s    | 1573.3           | 0.51 | 0.018 | 2000s   | 4690.7            | 0.54 | 0.015 |
> | ODA-B    | 500s    | 2032.6           | 0.76 | 0.009 | 2000s   | 5012.0            | 0.76 | 0.006 | 500s    | 888.9            | 0.64 | 0.021 | 2000s   | 2723.6            | 0.67 | 0.014 |
> | ODA-K    | 500s    | 2297.0           | 0.55 | 0.072 | 2000s   | 4596.2            | 0.63 | 0.050 | 500s    | 1074.5           | 0.22 | 0.053 | 2000s   | 4290.4            | 0.28 | 0.048 |
> | MOEAD    | 1000s   | 1488.3           | 0.61 | 0.059 | 3000s   | 3191.7            | 0.74 | 0.013 | 1000s   | 1724.0           | 0.48 | 0.023 | 3000s   | 2390.1            | 0.51 | 0.017 |
> | NAGAII   | 1000s   | 846.3            | 0.64 | 0.024 | 3000s   | 1105.7            | 0.68 | 0.022 | 1000s   | 819.6            | 0.28 | 0.036 | 3000s   | 1060.7            | 0.30 | 0.029 |
> | NAGAIII  | 1000s   | 185.4            | 0.67 | 0.025 | 3000s   | 291.4             | 0.70 | 0.014 | 1000s   | 299.5            | 0.43 | 0.031 | 3000s   | 427.9             | 0.50 | 0.025 |
> | UNSGAIII | 1000s   | 227.2            | 0.63 | 0.036 | 3000s   | 350.9             | 0.69 | 0.019 | 1000s   | 204.8            | 0.45 | 0.033 | 3000s   | 531.5             | 0.49 | 0.028 |
> | PMOCO    | 500s    | 325.9            | 0.74 | 0.020 | 2000s   | 880.5             | 0.75 | 0.012 | 500s    | -                | -    | -     | 2000s   | -                 | -    | -     |
> | Ours     | 500s    | 3308.5           | 0.75 | 0.010 | 2000s   | **6806.2**            | **0.77** | **0.005** | 500s    | **1823.2**           | **0.64** | **0.012** | 2000s   | **8270.8**            | **0.69** | **0.009** |
>
> **Q4) Regarding the sparse solutions by PMOCO.**
>
> We would like to note that in our testing of PMOCO, we find it is prone to generate the same points in objective space even with different preferences. Meanwhile, a large part of the generated points are dominated. In our experiments, we delete the dominated points and only keep the non-repetitive nondominated points, after we run PMOCO with all preferences. For example, we use 3654 preferences for each MOKP(4-100) instance as input to PMOCO, but the average number of nondominated points for each instance is only about 325.9. In the case where we use 7140 preferences, the average number of nondominated points is about 632. In summary, the number of finally derived nondominated points could be far less than the number of preferences.
>
> On the other hand, our graph learning based ODA guarantees that the solutions derived from the solver are nondominated by each other in the objective space and exactly in the true Pareto front. Therefore, in general, our method generates more nondominated points with better performance than the baselines.

---

> ### Author Response · Authors · 2022-08-02
> **Response to Reviewer LVBy (1/3)**
>
> We thank the reviewer for the valuable and positive comments.
>
> **Q1) Regarding the requirement of ODAs to collect data and imperfect dataset for behavioral cloning.**
>
> We would like to note that designing a totally brand new algorithm for solving MOIP with deep learning is not easy. While some learning based methods like PMOCO can realize the end-to-end learning, it is only applicable to multi-objective combinatorial optimization with sequential problem structure. Regarding the MOIP for general multi-objective combinatorial optimization, it is hard to design a neural network for solving all problems in an end–to-end manner, and a better choice is to deploy deep learning methods on top of mature algorithmic paradigms for MOIP, so that we can improve their performance for almost any problem. In fact, it is also common in the learning-to-optimize community to solve single-objective integer programming by improving branching or node selection in mature solvers like Gurobi or CPLEX. In this paper, we choose ODAs to deploy the graph learning, since they are the main stream to solve MOIP with mature algorithms, e.g., ODA-T/B/K.
>
> We acknowledge that given a time limit for solving MOIP, deploying the imitation learning on top of an algorithm with a better bounded-time performance than ODAs could be favorable. However, since our target is to solve general MOIP problems, the available algorithmic paradigms are still limited. The commonly used ones include ODAs, epsilon-constraint methods and branch & bound (or its variants), where ODAs are generally more efficient than the other two [1]. Therefore, we choose to deploy the graph learning on a given ODA. While the ODA is not a perfect teacher in bounded time, we empirically show that the ODA assisted by our graph learning can achieve a sufficiently good bounded-time performance, which already can outperform other heuristic baselines in most cases. Lastly, inspired by the comment of the reviewer, it would be meaningful to deploy our method on top of an ODA algorithm with the best performance in a time limit (such as among an ensemble of ODAs), so that the neural network can imitate better behaviors, and improve the eventual performance with such ensemble learning.
>
> **Q2) Regarding the use of RL for training the policy.**
>
> Reinforcement learning could be a feasible option to train our GNN, since it saves the efforts to prepare the labeled dataset and can search policies automatically. We also observe that some RL algorithms are designed to allow an agent to be time-aware and thus maximize the total reward in bounded time like the one in [2].  It is interesting to apply the related  RL techniques to train policies for solving MOIP.  However, how to design a proper RL to practically learn a favorable policy is not an easy task. Typically, it is much harder to design and tune a RL  algorithm to fully unleash its potential, which is more complex than training with imitation learning. Please kindly refer to our general response to reviewers for our views on possible issues by using RL and the reasons why we choose IL.
>
> Another potentially good training method is combining imitation learning with reinforcement learning. For example, the trained policy by imitation learning could be used as an initialized policy for reinforcement learning. In doing so, we can avoid training a RL policy from scratch with a random one, which generally consumes a long time to explore before finding a good policy. We will investigate this potential in future work.

---

### Official Review · Reviewer_A1Sc · 2022-07-11

**Rating:** 5
**Confidence:** 2
**Soundness:** 2 fair
**Presentation:** 2 fair
**Contribution:** 2 fair

**Summary:**

This study proposes a learning-based solver for effectively solving multi-objective integer programming (MOIP) by improving the bounded-time performance of objective-space decomposition algorithms (ODAs). Existing ODA has to solve large amounts of IPs and it incurs unnecessary and redundant computation. To reduce unnecessary computation, the current study proposed Markov Decision Process (MDP) framework for modeling the reduction procedure and learning the optimum reduction policy using the graph representation learning-based state representation.

**Questions:**

1. The definition of the state and action sounds make sense; however, it is not clear how the reward can be defined. Is there any sequential effect in the ODA solving process? For example, the decision at the current step will have a compounding long-term effect in obtaining the final solution for MOIP?

2. In line 228 of page 6, the paper says “We sample the state-action data in ODAs by using the optimal policy in Definition 3 as the oracle and then train the policy network in a supervised fashion”. Was the optimal policy is derived by solving the MDP formulated or the optimum policy can be always simply induced?

3. What was the motivation for employing imitation learning instead of reinforcement learning? Since the paper presented the MDP formulation (although it is not complete), is it also possible to employ Reinforcement learning?

4. The imitation learning forces the GNN policy to overfit to optimum action (labels), which can possibly limit the generalization capability compared to the RL method inherently conducting excessive exploration. What do you think where the power of generalization capability comes from?

5. How to choose a local upper point that will be discarded or retained by the learned policy? It seems that the set of nondominated points matters while the sequence for selecting them is not important in determining the next candidate solution. If it is true, how to impose such permutation invariance in the selected nondominated points?


**Limitations:**

•	It seems that ODA, one of the methods of solving the MOIP problem, has learned the policy to imitate the problem-solving method, but it did not clearly suggest how the presented method improved the performance and computation speed of the solution rather than just using ODA.

•	In order to apply imitation learning, it is necessary to obtain labeled data by optimally solving various problems. There are no experiments on whether there are any difficulties in obtaining the corresponding data, and how the performance changes depending on the size of the labeled data.


**Strengths And Weaknesses:**

<Strength>

It deals with the multi-objective IP problem that has not been dealt with relatively well in the field of artificial intelligence.


<Weakness>

The definition of MDP for solving the process of ODAs is not clear. The definition of the state and action sounds make sense; however, it is not clear how the reward can be defined. Is there any sequential effect in the ODA solving process? For example, the decision at the current step will have a compounding long-term effect in obtaining the final solution for MOIP?

It deals with the multi-objective IP problem that has not been dealt with relatively well in the field of artificial intelligence.

---

> ### Author Response · Authors · 2022-08-02
> **Response to Reviewer A1Sc (3/3)**
>
> **L2) Regarding the difficulty for data collection in IL and influence of dataset sizes.**
>
> We would like to note that the dataset collection is not a difficult task in our method. In fact, we can readily gather the dataset in a relatively short time. Please refer to our general response to reviewers.
>
> Furthermore, as a very early attempt to solve MOIP with graph learning, this paper focuses more on the effectiveness of the learned policy and the proposed GNN. Hence we conduct the ablation studies primarily on these two aspects, as did in most related literature [1] [5] [6]. Finding a more proper size of dataset or even selecting more proper training instances to boost the performance could also be an interesting future direction to investigate. However, here we quickly conduct an experiment to evaluate the influence of the size on the performance of our method. Given the size of the dataset collected with 1000s for each instance, i.e., about 8000 samples for each MOKP(3-100) instance and 5000 samples for MOAP(3-50), we use the 50%, 70%, 90% and 100% (i.e. our original method) of samples for each instance to retrain separate policies, respectively, and then test them with 10 instances and 1000s. We also collect 50% more samples (thus 150% in total) for each instance to train a policy and test the same instances with 1000s. We present the results as below. We observe that as the labeled dataset enlarges from size-50% to size-100%, the accuracy, cardinality (i.e. the number of nondominated points) and hypervolume increase. The policy trained by 100% samples (i.e. the one used in our paper) delivers the maximum hypervolume and cardinality for each of MOKP and MOAP, which suggests that our GNN can learn better reduction rules with sufficient samples. With more samples collected for training (size-150%), we find the learned policy slightly degrades on the hypervolume and cardinality compared to that of size-100%. It is reasonable since the policy is trained with samples beyond 1000s, which might harm the testing performance with only 1000s. Hence we advise that it might be a good practice to keep the time limit in testing the same as the one used in training. Despite the above (quick and rough) evaluation, we will conduct more comprehensive experiments to test the influence of dataset size in future.
>
> |              |       | MOKP(3-100) |       |       | MOAP(3-50) |       |
> |--------------|-------|-------------|-------|-------|------------|-------|
> | Dataset size | Accuracy | Cardinality       | HV    | Accuracy | Cardinality      | HV    |
> | size-50%     | 79.5  | 3291.1      | 0.672 | 81.0  | 3191.5     | 0.697 |
> | size-70%     | 81.6  | 3351.4      | 0.673 | 81.8  | 3353.6     | 0.699 |
> | size-90%     | 81.8  | 3562.0      | 0.673 | 81.9  | 3371.8     | 0.700 |
> | size-100%    | 82.8  | 3639.3      | 0.675 | 82.8  | 3500.3     | 0.708 |
> | size-150%    | 83.1  | 3580.9      | 0.673 | 83.4  | 3482.7     | 0.707 |
>
>
> Reference:
>
> [1] Maxime Gasse, Didier Chételat, Nicola Ferroni, Laurent Charlin, and Andrea Lodi. Exact combinatorial optimization with graph convolutional neural networks. In Proceedings of the 33rd Conference on Neural Information Processing Systems (NeurIPS), 2019.
>
> [2] Todd Hester, Matej Vecerik, Olivier Pietquin, et al. Deep q-learning from demonstrations. In Proceedings of the 32nd Conference on Artificial Intelligence (AAAI), 2018.
>
> [3] Zarpellon Giulia, Jason Jo, Andrea Lodi, and Yoshua Bengio. Parameterizing branch-and-bound search trees to learn branching policies. In Proceedings of the 35th Conference on Artificial Intelligence (AAAI), 2021.
>
> [4] Satya Tamby and Daniel Vanderpooten. Enumeration of the nondominated set of multiobjective discrete optimization problems. INFORMS Journal on Computing, 33(1):72–85, 2021.
>
> [5] Elias B Khalil, Christopher Morris, and Andrea Lodi. Mip-gnn: A data-driven framework for guiding combinatorial solvers. In Proceedings of the 36th Conference on Artificial Intelligence (AAAI), 2022.
>
> [6] Jianya Ding, Chao Zhang, Lei Shen, Shengyin Li, Bing Wang, Yinghui Xu, and Le Song. Accelerating primal solution findings for mixed integer programs based on solution prediction. In Proceedings of the 34th AAAI Conference on Artificial Intelligence (AAAI), 2020.

---

> > ### Comment · Reviewer_A1Sc · 2022-08-06
> > **Thank you for the reviewer's response.**
> >
> > Thank you for the reviewer's responses. The responses partially resolve my concerns.
> >
> > I believe reinforcement learning and imitation learning are not simply selected for learning effectiveness. Instead, reinforcement learning is learning a policy to choose an efficient instantaneous action to achieve a long-term goal under the MDP definition. In contrast, imitation learning is just imitating the best action at the current moment.
> >
> >  Is the action that imitation learning wants to imitate really an action that increases long-term performance, or is it a label that can be easily obtained by intuitive intuition? If the latter, I think this problem does not need to be defined as an MDP. The author's comment, "However, the long-term effect does not harm the performance of our method," makes me believe that the target problem does not have to be formulated as an MDP.
> >
> > The other critical difference between RL and IL is that RL can explore and thus find the better action used during training. However, imitation learning's performance is bounded by its labeled data. Then the benefit of the current study seems to be amortizing the optimum action computation using the learned neural network. If it does, the effectiveness of the proposed research should be measured by the reduction in the computation time by this amortizing and the performance reduction compared to the optimum decomposing action.

---

> > > ### Author Response · Authors · 2022-08-07
> > > **Response to Reviewer A1Sc (2/2)**
> > >
> > > **Q4) Regarding comparison with the optimal policy.**
> > >
> > > We would like to note that whether comparing with the (non-learning) optimal policy or not depends on the problem. For example, in the work [3] which aims to imitate a commonly used branching rule “strong branching”, the experiment shows comparison with such expert/teacher since it is a default rule in SCIP solver for practical use. In contrast, the defined (non-learning) optimal policy in our paper is just for the aim of data collection. In our implementation, we run ODA–T to solve instances to label the futile IPs with 1 and the others with 0, and the collected action-label pairs reflect an optimal policy in Definition 3 (in other words, if we use the labels as actions to solve the instances again, solving IPs by the solver will derive the exact Pareto front, without solving any futile IPs ). However, such optimal policy has less practical use, since it is meaningless to first run ODA-T to solve instances and then use the collected labels to solve them again. Instead, we endeavor to compare with typical and mature algorithms for MOIP to show the effectiveness of our method.
> > >
> > > However, to provide a glimpse of the performance of the (non-learning) optimal policy, we quickly test it on 10 MOKP(3-100) instances. Specifically, we first collect state-label pairs by solving instances with ODA-T, and then use the labels as actions for the optimal policy to solve the instances again. We only record the time for solving each instance with the optimal policy (excluding the time to collect the state-label pairs). We observe that the optimal policy finds all nondominated points for the 10 instances with 372.11s on average. For ODA-T with its own reduction rule, the time is 993.3s on average. For the learned reduction rule, it gains 81.1% nondominated points with 486.9s on average (note: the optimal policy is faster than the learned one, i.e. 372.11s vs 486.9s, since in each iteration, given the state-label pairs, the optimal policy saves the feature extraction and the computation through GNN to derive actions). It indicates that the optimal policy can achieve better (long-term) performance than the other policies, in terms of computation time or number of nondominated points. However, since it is only an theorectically ideal policy which has less practical use as mentioned above, please understand that we leave our main comparison with existing algorithms or GNNs, in terms of testing, generalization performance and ablation studies.
> > >
> > > Reference:
> > >
> > > [1] Luc Le Mero, Dewei Yi, Mehrdad Dianati, and Alexandros Mouzakitis. A survey on imitation learning techniques for end-to-end autonomous vehicles. IEEE Transactions on Intelligent Transportation Systems, 2022.
> > >
> > > [2] Edward Johns. Back to reality for imitation learning. In Proceedings of the 5th Annual Conference on Robot Learning (CoRL), 2022.
> > >
> > > [3] Maxime Gasse, Didier Chételat, Nicola Ferroni, Laurent Charlin, and Andrea Lodi. Exact combinatorial optimization with graph convolutional neural networks. In Proceedings of the 33rd Conference on Neural Information Processing Systems (NeurIPS), 2019.
> > >
> > > [4] Zarpellon Giulia, Jason Jo, Andrea Lodi, and Yoshua Bengio. Parameterizing branch-and-bound search trees to learn branching policies. In Proceedings of the 35th Conference on Artificial Intelligence (AAAI), 2021.
> > >
> > > [5] Jialin Song, Ravi Lanka, Yisong Yue, and Bistra Dilkina. A general large neighborhood search framework for solving integer linear programs. In Proceedings of the 34th Conference on Neural Information Processing Systems (NeurIPS), 2020.
> > >
> > > [6] He He, Hal Daume III, and Jason M Eisner. Learning to search in branch and bound algorithms. In Proceedings of the 27th Conference on Neural Information Processing Systems (NeurIPS), 2014.
> > >
> > > [7] Wen Sun, Anirudh Vemula, Byron Boots, and Drew Bagnell. Provably efficient imitation learning from observation alone. In International conference on machine learning (ICML), 2019.

---

> > > > ### Comment · Reviewer_A1Sc · 2022-08-08
> > > > **Thank you for your response.**
> > > >
> > > > Thank you for answering my questions.
> > > >
> > > > (1) IL can be considered as one of many ways to solve MDP in case IL can imitate the optimum action (in terms of the long-term accumulated return) given a state. However, I could not find that the action used in this study is indeed such a long-term best optimum action.  I believe this study uses MDP formulation to structure a decision-making policy; it temporally factorizes the entire sequential decision-making procedure using a stationary one-step decision-making policy. This is a way to express the target policy rather than a way to solve the target problem.
> > > >
> > > > (2) Thank you for the comparison between the imitation policy and the target non-learning decision-making rule.
> > > >
> > > > I believe properly armotizing well-known non-learning but effective decision-making routines will always improve computational efficiency and generalize to unseen problems. I think this paper empirically verifies that this idea can be applied to MOIP as well.
> > > > I hope the author includes the above discussion in the revised manuscript.
> > > >
> > > > I will change my score to "borderline accept"

---

> > > > > ### Author Response · Authors · 2022-08-09
> > > > > **Response to Reviewer A1Sc**
> > > > >
> > > > > We greatly appreciate the reviewer for the acknowledgement of our idea and the experimental results. Here, we try to explain a bit more about the new comments.
> > > > >
> > > > > **Q1) Regarding the optimal policy in our paper.**
> > > > >
> > > > > We would like to note that our MDP and the policy is defined on top of an ODA. Except the action (i.e. discarding an IP or solving it), all other algorithmic parts remain unchanged (e.g. the rule to select IP). A policy is defined as optimal, if it decides to call the solver to solve the IPs which contain all non-repetitive nondominated points, and discard (i.e. not solve) all futile IPs. “Optimal” emphasizes the number of solver calls exactly equals to the cardinality of Pareto front, without any invalid solving that causes unnecessary runtime (compared to existing reduction rules or the learned ones), as stated in Definition 3 in our paper.
> > > > >
> > > > > Given a fixed ODA (ODA-T in our paper), our defined optimal policy can gain very good long-term performance within the ODA, since it achieves all nondominated points in a short time. For example, the defined optimal policy takes possible actions 0 (solve), 0 (solve), 1 (discard), 1 (discard), 0 (solve) for an instance, with all three nondominated points in Pareto front attained.  A stochastic policy may take actions 0 (solve), 0 (solve), 0 (solve), 1 (discard), 1 (discard), which means it computes the futile IP in the 3rd step (with extra runtime) and discard the nondominated point in the last step. In fact, the defined optimal policy is better than existing reduction rules in ODA which often induce the solving of futile IPs.
> > > > >
> > > > > Given the above favorable long-term performance, our defined optimal policy is fairly good for IL to imitate and thus attains promising empirical results, which show our method can generally achieve more nondominated points than the baselines in a short runtime.
> > > > >
> > > > > **Q2) Regarding MDP and IL solution.**
> > > > >
> > > > > We would like to note that it is natural to describe a sequential decision making process by MDP, when the process satisfies the Markov property and captures the interaction between agent decision and environment. Given the optimization objective to maximize the cumulative reward, RL is commonly used to learn the decisions. However, IL is a good alternative if the rewards cannot be readily achieved or designed.
> > > > >
> > > > > We acknowledge that IL does not explicitly optimize the objective in the training loss, as did by RL. However, IL (as an alternative to RL) without the need of explicit rewards, could produce a good solution to the MDP objective by trying to imitate behaviors (i.e. learn from demos) of a good policy, which yield high long-term return. This might also be the reason why IL has been widely used for sequential decision making problems and its effectiveness in different scenarios has also been verified.
> > > > > *******************
> > > > > According to the advice of the reviewer, we will include a section to discuss those points in the final version. We sincerely thank the reviewer again for raising the score.

---

> > > ### Author Response · Authors · 2022-08-07
> > > **Response to Reviewer A1Sc (1/2)**
> > >
> > > We thank for the reviewer's valuable comments.
> > >
> > > **Q1) Regarding selecting IL rather than RL (further explanation).**
> > >
> > > We would like to note that imitation learning (IL) has been widely studied and used in sequential decision making problems in domains of autonomous driving [1], robot learning [2], etc., especially integer programming in recent years [3-6]. It aims to learn from demos (i.e. sequences of actions) which typically have good (or even optimal) sequential performance. Compared to RL, which maximizes the long-term return (i.e. cumulative reward) by capturing the future reward in the update of the current decision, the logic of IL is directly imitating the favorable policy which gains high long-term return, by cloning the trajectory distribution of the policy given initial states. In other words, IL also considers long-term effect in the sequential decision making, in the form of demos from the favorable policy.
> > >
> > > We also note that there is no absolute advantage of IL or RL over each other. In our view, IL can be regarded as an important alternative to RL.
> > >
> > > We do acknowledge that RL can explore actions to learn policy. However, its efficacy may severely rely on an elegant reward function and a proper exploration strategy. In our problem, designing such rewards and strategy is nontrivial under the long episodes with large state space, and the naive design might induce inferior policy training. In contrast, IL focuses on learning from the (non-learning) optimal policy with favorable long-term return. It constrains the training to a small but promising state space, which may largely reduce the training complexity, and meanwhile bypasses the trial and error for reward designing/reshaping. While we do acknowledge that the performance of IL is bounded by the labeled data, the labeled data themselves in our paper are optimal since they are obtained from the (non-learning) optimal policy, which  might help accelerate the training for a good learned policy.
> > >
> > > Finally, we do not deny that RL may find better policies than the one learned by IL. We believe that it has good potential if effective reward function and exploration strategy are well designed, which is beyond the scope of this paper. We do hope our work in this paper (as the first learning based method for MOIP) can inspire such RL works.
> > >
> > >
> > > **Q2) Regarding the necessity of MDP formulation.**
> > >
> > > According to the Definition 3, the (non-learning) optimal policy can always delete futile IPs (that lead to infeasibility and repetitive nondominated points) and solve IPs which contain the exact Pareto front. It corresponds to actions that increase long-term performance since,
> > >
> > > (1) compared to existing reduction rules in ODA-T/B/K, which correspond to policies with frequent actions for solving futile IPs by the solver, the optimal policy avoids solving such IPs and thus largely reduces the computation time in solving each instance (from RL perspective, if we define negative rewards for invalid solving, the optimal policy will avoid all these rewards and achieve higher long-term return);
> > >
> > > (2) compared to the learned policies (i.e. learned reduction rules) which could wrongly delete IPs with nondominated points, the optimal policy will derive the whole Pareto front when the instance is exactly solved (from RL perspective, if we define positive reward for finding a new nondominated point, the optimal policy can achieve all these positive rewards and thus higher long-term return).
> > >
> > > In other words, different policies could induce different MDPs with various long-term performance. Hence, it is proper to describe the solving process with MDP.
> > >
> > > Meanwhile, the ODA solving process satisfies the nature of MDP. In other words, the current state is only determined by the previous state (the previous layout of local upper bounds, found nondominated points, etc.) and the action (discarding the current IP or not). The use of MDP benefits the statement of our method and enhances the readability of the paper. Hence, keeping the MDP formulation could be more reasonable.
> > >
> > >
> > > **Q3) Regarding our comment.**
> > >
> > > Our comment "However, the long-term effect does not harm the performance of our method," aims to describe that, extensive experimental results show the learned policy by our method suffers less from the compounding errors given the reported good performance, while sometimes the compounding errors may harm the use of IL policy in general. It does not mean that our problem has no long-term effect or does not have to be formulated as an MDP. In fact, IL (as an alternative to RL) is often used for sequential decision making, which sometimes could be described by MDP [3] [5][7].

---

> ### Author Response · Authors · 2022-08-02
> **Response to Reviewer A1Sc (2/3)**
>
> **Q4) Regarding the generalization of IL.**
>
> The results in our paper have verified good generalization capacities of the proposed method across problem sizes and objective numbers.  The generalization may come from, 1) the novel two-stage GNN which can learn effective high-level representations of MOIP to discriminate the futile IPs; 2) the good featurization of MOIP, which is less sensitive to problem sizes or the number of objectives. Given proper features and a powerful GNN, the learned policy by imitation learning has favorable potential to generalize well across problem sizes, as also empirically verified in literature, e.g., [1] [3]. On the other hand, we also leverage early stopping in our training which is a commonly used technique to further avoid overfitting (line 272 in page 7 of the paper).
>
> We acknowledge that RL could have a good generalization performance when the agent interacts with the environment sufficiently. However, as we mentioned in the general response, the exploration itself is a hard issue, especially in the context of MOIP with relatively long episodes and extremely diverse MDPs. How to encourage the agent to wisely and efficiently explore and avoid the early convergence to an inferior local optimum is not an easy task.
>
> **Q5) Regarding the selection of local upper bound points.**
>
> The selection of local upper bounds is another important component in an ODA algorithm. Without any prior information, it is intuitive for an ODA algorithm to select a local upper bound which is more likely to contain a nondominated point. Hence, a heuristic rule could select the local upper bound with the maximum unexplored part in objective space, as did in ODA-T [4] which we use to deploy the graph learning method. We keep this selection rule unchanged in our method (i.e. regard it as the environment in MDP), since we only focus on improving the reduction rule.
>
> We also acknowledge that designing more effective selection rules are expected to further improve the performance of the ODA, e.g., controlling the layout (in the objective space) of nondominated points generated in bounded time. It is also interesting to learn such rules with our graph learning based method, which however is beyond the scope of this paper.
>
> **L1) Regarding how our graph learning method improves the original ODA.**
>
> ODA algorithms normally decompose MOIP into a sequence of single-objective IPs, which are then solved by a solver. In each iteration, the most runtime is spent on solving an IP. However, existing ODAs generally have no effective reduction rules to filter futile IPs (that cause infeasibility or repetitive nondominated points). Specifically, ODAs generally cannot identify such futile IPs in prior, and have to solve them by a solver to determine whether they are futile or not, which induces unnecessary or fruitless runtime along the solving process. In contrast, our method aims to leverage a GNN to learn an effective reduction rule for an ODA to predict whether an IP is futile before solving it. If the prediction is “yes”, we will discard the IP, which saves the time to solve it. Given a bounded time, the ODA could spend more time to focus on solving fruitful IPs with nondominated points, so that more effective points can be found. Please refer to our motivation and the detailed analysis on reduction rules in Section 1 and Section 4.1, respectively.
>
> As analyzed in Remark in Section 4.1, our method essentially relaxes the optimality guarantee of the ODA, since the neural network could wrongly discard some valid IPs with nondominated points in the final Pareto front. However, given bounded runtime, our learning assisted method can accelerate the search of more nondominated points, which increase the bounded performance, as shown in the experimental results. For example, from Table 1 and 2 in our paper, it is clear that our method (for 7 out of 8 testing instance groups) gains more nondominated points than the original ODA (i.e. ODA-T) and achieves significantly better performance in terms of HV and IGD.

---

> ### Author Response · Authors · 2022-08-02
> **Response to Reviewer A1Sc (1/3)**
>
> We appreciate the reviewer for the valuable and constructive comments.
>
> **Q1) Regarding the reward definition and sequential effect.**
>
> We skip this reward description in our paper as did in [1], since we adopt imitation learning which does not need the reward in its training. However, we agree that an explicit reward description may make the MDP more complete and informative. Given our goal to increase the number of nondominated points attained in bounded time, the reward could be intuitively set to a positive value (e.g. 1) if the current IP is solved to gain a new nondominated point, and set to 0 otherwise if the current IP is discarded or solved without gaining a new nondominated point. Also, a discount factor smaller than 1 is necessary to encourage the RL agent to emphasize the early reward, so that it could be good at maximizing the total reward within the time limit (i.e. the number of nondominated points in bounded time).
>
> We would like to note that the current decision directly influences the final solution in two aspects. On the one hand, wrong decisions may cause less nondominated points in the final solution, e.g. by wrongly discarding IPs with non-repetitive nondominated points, or wrongly solving futile IPs in the bounded time so that less time is available to search valid nondominated points. On the other hand, the decision influences the update of local upper bounds, which thus influences the states and the sequence of IPs that are selected from the local upper bounds. The above aspects together would exert a long-term effect on the final solution to MOIP.  However, the long-term effect does not harm the performance of our method. Extensive results in our paper have shown that the learned policy can generally identify futile IPs accurately to gain far more nondominated points than all baselines, and it can also well generalize to different instances to tackle different sequences of IPs.
>
> **Q2) Regarding how the optimal policy is derived.**
>
> The Definition 3 in our paper presents a general description of the optimal policies. However, one such optimal policy used in our data collection can be simply derived.  Specifically, we only need to solve each training instance with an ODA algorithm and label the solved IPs (1 for futile IPs and 0  otherwise) along the solving process. If we use the collected labels to guide the ODA to solve a training instance again (i.e. labels means discarding the IP (1) or not (0)), all futile IPs will be avoided and the IPs solved will derive all nondominated points. Hence, the collected state-action pairs capture the behavior of an optimal policy as in Definition 3, which we use to train our policy. Please refer to Section 4.4 and pseudocode in Appendix A.5 for more details about the dataset collection.
>
> **Q3) Regarding the motivation for using IL over RL.**
>
> Please refer to our general response to reviewers for our views in choosing imitation learning (IL) over reinforcement learning (RL) for solving MOIP. In fact, to derive a better policy by RL than that of IL, it is  critical to tackle the inefficient exploration issue. Otherwise, the RL may take much longer time to learn a good policy. A potential alternative to bypass the above issue for RL might be that we could leverage the learned policy by IL as the initialized policy for RL, and then explore for a further better policy. In doing so, we can save samples and time in RL training to gain a competitive policy, since the IL agent is already able to achieve good performance (according to experimental results in our paper), and it is far better than a randomly initialized policy to start the exploration completely from scratch. We would like to note that similar ideas have been studied previously, e.g. [2]. Although it is beyond the scope of this paper, we will try to further improve the training algorithm in such fashion in future.

---

### Official Review · Reviewer_gmdK · 2022-07-12

**Rating:** 7
**Confidence:** 3
**Soundness:** 3 good
**Presentation:** 3 good
**Contribution:** 3 good

**Summary:**

The paper proposes a GNN-based method for objective space decomposition (ODA) in IP problems.
ODA is relevant when multiple objectives are to be optimized, but solvers usually only consider a single objective at a time.
The ODA method decides which objective to optimize next to best (closest and fastest) approximate the true Pareto front.
The new method models the IP problem as a graph and applies a GNN to classify which objective to optimize next. The network is trained using behavioral cloning, i.e. supervised learning.
Experiments show that the new method is competitive to or outperforming existing ODA techniques and evolutionary multi-objective algorithms.

**Questions:**

1. Can you elaborate on the tradeoffs for collecting the dataset? Do you think an online learning approach would be competitive?
2. Why do you fix the instance sizes for training?
3. What would it take to generalize the model further, i.e. being problem-independent and learning from a joint dataset for MOKP+MOAP+other problems?


**Limitations:**

Some limitations of the work are discussed (solving intractable problems) as well as the cases where the proposed method is outperformed by the comparison techniques.

**Strengths And Weaknesses:**

The method is strictly designed for the ODA approach to be solved, which allows it to gains the performance improvements, but does perhaps not leave much room for further generalization. However, since ODA is already a general problem, this should be okay.
The approach itself is well motivated and intuitive enough to understand, adopt and potentially reproduce.
An advantage is that the method is independent of the solver itself.
My main concern is on the training of the method which requires a dataset for supervised learning. I'd be curious to know more about the challenges and tradeoffs in building this dataset (it is briefly discussed, but it seems to be one of the main efforts in setting this up for practical use).

If I understood it correctly, the model is trained for a specific problem and instance size, e.g. MOKP with 3 objectives and 100 items.
Is it a limitation of the GNN that the size of the instances must be fixed? I understand it to some degree for the objectives, even though generalization is tested later, but one advantage of using GNNs should also be a more flexible structure in the instance sizes themselves.
While some generalization capabilities are shown, the method is, as described, mostly fixed to specific instance sizes. It is unclear if this is a necessity, a design decision, or so far the first step in generalizing the technique.

---

> ### Author Response · Authors · 2022-08-02
> **Response to Reviewer gmdK (2/2)**
>
> Reference:
>
> [1] Maxime Gasse, Didier Chételat, Nicola Ferroni, Laurent Charlin, and Andrea Lodi. Exact combinatorial optimization with graph convolutional neural networks. In Proceedings of the 33rd Conference on Neural Information Processing Systems (NeurIPS), 2019.
>
> [2] Jialin Song, Ravi Lanka, Yisong Yue, and Bistra Dilkina. A general large neighborhood search framework for solving integer linear programs. In Proceedings of the 34th Conference on Neural Information Processing Systems (NeurIPS), 2020.
>
> [3] Yeong-Dae Kwon, Jinho Choo, Byoungjip Kim, Iljoo Yoon, Youngjune Gwon, and Seung jai Min. Pomo: Policy optimization with multiple optima for reinforcement learning. In Proceedings of the 34th Conference on Neural Information Processing Systems (NeurIPS), 2020.
>
> [4] Yunhao Tang, Shipra Agrawal, and Yuri Faenza. Reinforcement learning for integer programming: Learning to cut. In Proceedings of the 37th International Conference on Machine Learning (ICML), 2020.
>
> [5] Iklassov Zangir, Dmitrii Medvedev, Ruben Solozabal, and Martin Takac. Learning to generalize Dispatching rules on the Job Shop Scheduling. arXiv preprint arXiv:2206.04423, 2020.
>
> [6] Yuan Jiang, Yaoxin Wu, Zhiguang Cao, Jie Zhang. Learning to Solve Routing Problems via Distributionally Robust Optimization. In Proceedings of the 36th AAAI Conference on Artificial Intelligence (AAAI), 2020.
>
> [7] Manchanda, Sahil, Sofia Michel, Darko Drakulic, and Jean-Marc Andreoli. On the Generalization of Neural Combinatorial Optimization Heuristics. arXiv preprint arXiv:2206.00787, 2022.

---

> ### Author Response · Authors · 2022-08-02
> **Response to Reviewer gmdK (1/2)**
>
> We thank the reviewer for the valuable and positive comments.
>
> **Q1) Regarding challenges and tradeoffs to build the dataset for supervised learning.**
>
> We would like to note that the dataset collection is not a particularly challenging issue in our method. Please refer to the general response to reviewers. Regarding the tradeoffs, we clarify that the time limit could be set empirically according to instances. Typically, the futile IPs (that cause infeasibility or repetitive nondominated points) will not appear uniformly along the solving process of the ODA. For example, we observe that the futile IPs are sparse at the early stage and then become relatively dense. Hence we empirically set the time limit to 1000s, where we could gather positive and negative samples without significant difference in quantity, which help avoid the issue of  imbalanced dataset and also benefit the policy training. Despite the desirable effectiveness of 1000s for our problems (MOKP and MOAP), we advise users to run a small number of instances by the ODA to observe the rough layout of futile IPs and then determine the time limit for bypassing the possible bias in data, when they aim to deploy the proposed method to solve new problems with different ODAs.
>
> **Q2) Regarding the online learning setting.**
>
> To our knowledge, online learning is good at updating the machine learning model with real-time feedback, which may avoid the potentially excessive use of memory in batch learning otherwise. While there is no such memory issue in our current experiments, we think that online learning could be a better choice than the batch learning, especially when problem scales become too large to train the learning model even with a moderate batch size. It would be very interesting to extend our method to an online learning setting for solving MOIPs,  which we will give a shot  in our future work.
>
> **Q3) Regarding the generalization to other sizes and fixed instance sizes for training.**
>
> The reviewer’s understanding is right. We train the model with a specific problem and instance size, and then generalize the trained model to other sizes. To our best knowledge, this training paradigm is broadly used in the learning-to-optimize literature including solving integer programming (and other general combinatorial optimization) [1] [2] [3] [4]. It is a standard way that the neural network is trained on instances with a fixed size, and then generalizes to different sizes. One of the motivations is that the problem features are more similar under the same size, which may benefit the policy training.  Then, given the scalability nature of GNN, the learned policy will gain the generalization capacity on instances of different sizes. Hence, a learned policy is often evaluated on instances from the distribution of training set and out-of-the-distribution instances with different sizes [1] [3] [4]. Despite the above basic generalization study, some techniques are specifically designed on top of the model to further enhance the generalization across sizes or distributions, e.g. training with instances of mixed sizes [5]. In other words, our method could be further improved for increasing the specific generalization capacity, which however is beyond the current scope of this paper.
>
> On the other hand, we would like to note that our GNN model can be used on varying problem sizes and even different numbers of objectives (note: it has not been realized in previous learning based methods). We have empirically verified the favorable cross-size and cross-objective generalization of the GNN in our paper. Furthermore, our model has a great potential to be used as a backbone to further boost the generalization capacity on MOIPs with specialized techniques.
>
> **Q4) Regarding generalizing the model further.**
>
> In fact, speaking of the generalization, we should consider which generalization capacity we want the model to gain. If we focus on cross-problem generalization, it is intuitive to train the model with different problems together, as indicated by the reviewer. For MOIPs, generalization across problem sizes and objective numbers are important in our views. Moreover, we could resort to some techniques in the learning-to-optimize literature (e.g. meta-learning) to further improve such generalization capacities [5] [6] [7].  No matter which generalization capacity, including the cross-problem one mentioned by the reviewer, our method with the novel two-stage GNN will stand as a good choice to develop the corresponding methods, since it is already able to work relatively well with different problem sizes and objective numbers, as shown in the  generalization study in our paper.

---

> > ### Comment · Reviewer_gmdK · 2022-08-05
> > **Response**
> >
> > Thank you for the detailed response to my comments. I'm positive about this work and keep my score as before.
> >
> > My questions were more on the general, method-design side and not necessarily a criticism of the work presented.
> > I agree mostly with the answers, and it is true that many aspects (e.g., using imitation learning or training on separated problem and sizes) is used commonly in the literature.
> > It should not be a negative point that it is done similarly when the focus is on something else. Still, while these are status quo at the moment, I have the impression that we should overcome them by some alternatives in the future, as also mentioned by the authors.

---

> > > ### Author Response · Authors · 2022-08-05
> > > **Response to Reviewer gmdK**
> > >
> > > Thanks for your acknowledgement and support.

---

### Author Response · Authors · 2022-08-02
**General Response to Reviewers**

We greatly appreciate the reviewers for reviewing the paper and offering valuable comments. Generally, the reviewers acknowledge the novelty of our method and significance of empirical results. They also agree that MOIP is a hard but important problem, which deserves more attention and study. In what follows, we firstly summarize and respond to two common questions from reviewers, and then present point-to-point responses to individual reviewers.

**Q1) Regarding replacing imitation learning (IL) with reinforcement learning (RL) for training** ( Reviewer A1Sc and Reviewer LVBy)

Given a carefully designed reward, RL can be used to maximize the total expected reward that relates to the objective function, without the need of preparing the labeled dataset as required in IL. However, searching for a competitive policy by RL often costs more samples and longer time.

For our problems, we note that the RL agent has to explore a large state space, i.e., long episodes for each instance (thousands of steps) with varying states. Given a randomly initialized policy, it is generally inefficient for RL to improve the policy towards a good one. A large number of instances would be required for convergence with a good performance, which means long training time (especially considering 1000s time limit for each instance).  Furthermore, hyperparameters in RL have to be carefully tuned, e.g. discount factor, epsilon or entropy for exploration. Also, reward shaping is often needed but tricky for RL to avoid locally optimal policies. A discussion on problems of RL for solving integer programming is also given in Section 4 of literature [1].

On the other hand, given that our aim is to discriminate or identify futile IPs, it is more natural to regard it as a classification task. Thus we opt for imitation learning, which has also been justified in the previous studies for single-objective integer programming [1] [2] [3] [4].  Extensive results have shown that the learned policy based on IL achieves steadily good performance, in terms of testing and generalization.

**Q2) Regarding the challenge (or difficulty) to collect dataset** (Reviewer gmdK and Reviewer A1Sc)

The dataset collection is relatively easy in our method. Specifically, we run an ODA algorithm with the chosen time limit (1000s in our paper) to solve each training instance, and label IPs (1 for futile ones and 0  otherwise) along the solving process, as stated in Section 4.4 and the pseudocode is given in Appendix A.5. In practice, one can also run instances in parallel with CPUs. In our implementation, we divide 100 training instances into 3 groups and collect the dataset with 3 CPUs in parallel. It costs about 8h to gather the entire dataset, which is satisfactorily acceptable.

Reference:

[1] Maxime Gasse, Didier Chételat, Nicola Ferroni, Laurent Charlin, and Andrea Lodi. Exact combinatorial optimization with graph convolutional neural networks. In Proceedings of the 33rd Conference on Neural Information Processing Systems (NeurIPS), 2019.

[2] He He, Hal Daume III, and Jason M Eisner. Learning to search in branch and bound algorithms. In Proceedings of the 27th Conference on Neural Information Processing Systems (NeurIPS), 2014.

[3] Jialin Song, Ravi Lanka, Yisong Yue, and Bistra Dilkina. A general large neighborhood search framework for solving integer linear programs. In Proceedings of the 34th Conference on Neural Information Processing Systems (NeurIPS), 2020.

[4] Vinod Nair, Sergey Bartunov, Felix Gimeno, Ingrid von Glehn, Pawel Lichocki, Ivan Lobov,Brendan O’Donoghue, Nicolas Sonnerat, Christian Tjandraatmadja, Pengming Wang, et al. Solving mixed integer programs using neural networks. arXiv preprint arXiv:2012.13349, 2020.

---

### Comment · Area_Chair_mRhZ · 2022-08-06
**Please read the authors' response and start the discussion**

Dear Reviewers,

Thanks for providing the review. The discussion stage will end in next Tuesday. Please check the authors' response and feel free to discuss with authors.

Best, AC

---

### Meta-Review · Area_Chair_mRhZ · 2022-08-29

**Recommendation:** Accept
**Confidence:** Less certain

**Metareview:**


In this paper, the authors exploit imitation learning with a two-stage GNN to learn the reduction rule for ODA to accelerate the solver and reduce the unnecessary computation. The authors evaluate the performances of the proposed method and demonstrate the advantages.


In sum, this paper consider an interesting application of machine learning for optimization and provide a promising solution. All reviewers provide relatively positive feedback of this submission.

Please consider the reviewers' suggestions to improve the submission:

- Justify the MDP modeling with concrete definition of the state and action for the reduction rule for ODA.

- Specify the data set construction and justify the generalization ability in the imitation learning.
- Provide comprehensive comparison, especially with PMOCO.


**Award:**

No

---

### Decision · Program_Chairs · 2022-09-14

Accept